# SUPER-ADAM: Faster and Universal Framework of Adaptive Gradients

**Feihu Huang, Junyi Li, Heng Huang**

Department of Electrical and Computer Engineering, University of Pittsburgh, Pittsburgh, USA

`huangfeihu2018@gmail.com, junyili.ai@gmail.com, heng.huang@pitt.edu`

## Abstract

Adaptive gradient methods have shown excellent performances for solving many machine learning problems. Although multiple adaptive gradient methods were recently studied, they mainly focus on either empirical or theoretical aspects and also only work for specific problems by using some specific adaptive learning rates. Thus, it is desired to design a universal framework for practical algorithms of adaptive gradients with theoretical guarantee to solve general problems. To fill this gap, we propose a faster and universal framework of adaptive gradients (i.e., SUPER-ADAM) by introducing a universal adaptive matrix that includes most existing adaptive gradient forms. Moreover, our framework can flexibly integrate the momentum and variance reduced techniques. In particular, our novel framework provides the convergence analysis support for adaptive gradient methods under the nonconvex setting. In theoretical analysis, we prove that our SUPER-ADAM algorithm can achieve the best known gradient (i.e., stochastic first-order oracle (SFO)) complexity of $\tilde{O}(\epsilon^{-3})$ for finding an $\epsilon$-stationary point of nonconvex optimization, which matches the lower bound for stochastic smooth nonconvex optimization. In numerical experiments, we employ various deep learning tasks to validate that our algorithm consistently outperforms the existing adaptive algorithms. Code is available at https://github.com/LIJUNYI95/SuperAdam

## 1 Introduction

In the paper, we consider solving the following stochastic optimization problem:

$$\min_{x \in \mathcal{X}} f(x) := \mathbb{E}_{\xi \sim \mathcal{D}}[f(x; \xi)], \tag{1}$$

where $f(x)$ denotes a smooth and possibly nonconvex loss function, and $\xi$ is a random example variable following an unknown data distribution $\mathcal{D}$. Here $\mathcal{X} = \mathbb{R}^d$ or $\mathcal{X} \subset \mathbb{R}^d$ is a compact and convex set. The problem (1) frequently appears in many machine learning applications such as the expectation loss minimization. Recently, Stochastic Gradient Descent (SGD) [14] is commonly used to solve the problem (1) such as Deep Neural Networks (DNNs) training [18, 20], due to only requiring a mini-batch samples or even one sample at each iteration. Adaptive gradient methods are one of the most important variants of SGD, which use adaptive learning rates and possibly incorporate momentum techniques, so they generally require less parameter tuning and enjoy faster convergence rate than SGD. Meanwhile, compared to SGD, adaptive gradient methods escape saddle points faster [31]. Thus, recently adaptive gradient methods have been widely developed and studied. For example, the first adaptive gradient method *i.e.*, Adagrad has been proposed in [12], which significantly outperforms the vanilla SGD under the sparse gradient setting. Subsequently, some variants of Adagrad *e.g.*, SC-Adagra [28] and SAdagrad [9] have been proposed for (strongly) convex optimization. Unfortunately, Adagrad has been found that it does not be well competent to the dense gradient setting and the nonconvex setting. To address this drawback, some other efficient variants of

35th Conference on Neural Information Processing Systems (NeurIPS 2021).

Table 1: Gradient (i.e., stochastic first-order oracle (SFO)) complexity of the representative adaptive gradient algorithms for finding an $\epsilon$-stationary point of the **non-convex** stochastic problem (1), i.e., $\mathbb{E}\|\nabla f(x)\| \leq \epsilon$ or its equivalent variants. For fair comparison, we only provide the gradient complexity and convergence rate in **the worst case** without considering the sparsity of stochastic gradient. Here $T$ denotes the whole number of iteration, and $b$ denotes mini-batch size. **ALR** denotes adaptive learning rate. **1** denotes the smoothness of each component function $f(x;\xi)$; **2** denotes the smoothness of objective function $f(x) = \mathbb{E}_\xi[f(x;\xi)]$; **3** denotes the bounded stochastic gradient $\nabla f(x;\xi)$; **4** denotes the bounded true gradient $\nabla f(x)$; **5** denotes that $f(x)$ is Lipschitz continuous; **6** denotes the smoothness of true gradient $\nabla f(x)$.

| Algorithm | Reference | Complexity | Convergence Rate | ALR | Conditions |
|---|---|---|---|---|---|
| Adam/ YOGI | [36] | $O(\epsilon^{-4})$ | $O(\frac{1}{\sqrt{T}} + \frac{1}{\sqrt{b}})$ | specific | **1, 2, 3, 4** |
| Generalized Adam | [8] | $\tilde{O}(\epsilon^{-4})$ | $O(\frac{\sqrt{\log(T)}}{T^{1/4}})$ | specific | **2, 3, 4** |
| Padam | [6] | $O(\epsilon^{-4})$ | $O(\frac{1}{\sqrt{T}} + \frac{1}{T^{1/4}})$ | specific | **2, 3, 4** |
| Adaptive SGD | [23] | $\tilde{O}(\epsilon^{-4})$ | $O(\frac{\sqrt{\ln(T)}}{\sqrt{T}} + \frac{\sqrt{\ln(T)}}{T^{1/4}})$ | specific | **2, 5** |
| AdaGrad-Norm | [34] | $\tilde{O}(\epsilon^{-4})$ | $O(\frac{\sqrt{\log(T)}}{T^{1/4}})$ | specific | **2, 4** |
| Ada-Norm-SGD | [10] | $\tilde{O}(\epsilon^{-3.5})$ | $\tilde{O}(\frac{1}{T^{2/7}})$ | specific | **2, 6** |
| AdaBelief | [40] | $\tilde{O}(\epsilon^{-4})$ | $O(\frac{\sqrt{\log(T)}}{T^{1/4}})$ | specific | **2, 3, 4** |
| Adam$^+$ | [25] | $O(\epsilon^{-3.5})$ | $O(\frac{1}{T^{2/7}})$ | specific | **2, 6** |
| STORM | [11] | $\tilde{O}(\epsilon^{-3})$ | $O(\frac{(\ln(T))^{3/4}}{\sqrt{T}} + \frac{\sqrt{\ln(T)}}{T^{1/3}})$ | specific | **1, 3, 4** |
| SUPER-ADAM ($\tau=0$) | Ours | $\tilde{O}(\epsilon^{-4})$ | $O(\frac{\sqrt{\ln(T)}}{\sqrt{T}} + \frac{\sqrt{\ln(T)}}{T^{1/4}})$ | universal | **2** |
| SUPER-ADAM ($\tau=1$) | Ours | $\tilde{O}(\epsilon^{-3})$ | $O(\frac{\sqrt{\ln(T)}}{\sqrt{T}} + \frac{\sqrt{\ln(T)}}{T^{1/3}})$ | universal | **1** |

Adagrad, *e.g.*, Adadelta [37], Adam [22], have been presented by using exponential moving average instead of the arithmetic average.

Adam [22] recently has been shown great successes in current machine learning problems, *e.g.*, it is a default method of choice for training DNNs [17] and contrastive learning [7]. Unfortunately, Reddi et al. [29] still showed that Adam is frequently divergent in some settings where the gradient information quickly disappear. To deal with this issue, some variants of Adam algorithm, *e.g.*, AMSGrad [29], YOGI [36] and generalized Adam [8] have been proposed. Specifically, AMSGrad [29] applies an extra 'long term memory' variable to preserve the past gradient information in order to handle the convergence issue of Adam. YOGI [36] introduces an adaptive denominator constant, and studies effect of the mini-batch size in its convergence. Subsequently, Chen et al. [8] studied the convergence of a class of Adam-type algorithms for nonconvex optimization. Zhou et al. [39] analyzed the convergence of a class of adaptive gradient algorithms for nonconvex optimization, and the result shows the advantage of adaptive gradient methods over SGD in sparse stochastic gradient setting. Meanwhile, Liu et al. [24] studied the variances of these adaptive algorithms. More recently, Guo et al. [19] presented a novel convergence analysis for a family of Adam-style methods (including Adam, AMSGrad, Adabound, etc.) with an increasing or large momentum parameter for the first-order moment.

Although the above these adaptive gradient methods show some good empirical performances, their generalization performance is worse than SGD (with momentum) on many deep learning tasks due to using the coordinate-wise learning rates [35]. Thus, recently some adaptive gradient methods have been proposed to improve the generalization performance of Adam. For example, AdamW [26] and Padam [6] improve the generalization performance of Adam by decoupling weight decay regularization and introducing a partial adaptive parameter, respectively. Luo et al. [27] proposed a new variant of Adam (*i.e.*, Adabound) by employing dynamic bounds on learning rates to improve the generalization performance. Subsequently, AdaBelief [40] has been presented to obtain a good generalization by adopting the stepsize according to the 'belief' in the current gradient direction. In addition, the norm version of AdaGrad (*i.e.*, AdaGrad-Norm) [34] has been proposed to obtain a good generalization performance.

So far, the above adaptive gradient methods still suffer from a high gradient complexity of $O(\epsilon^{-4})$ for finding $\epsilon$-stationary point in the worst case without considering sparsity of gradient. More recently, some faster variance-reduced adaptive gradient methods such as STORM [11], Adaptive Normalized

SGD [10], Adam$^+$ [25] have been proposed. For example, STORM applies the momentum-based variance reduced technique to obtain a lower gradient complexity of $\tilde{O}(\epsilon^{-3})$. To the best of our knowledge, all these existing adaptive gradient methods only use some specific adaptive learning rates with focusing on either pure theoretical or empirical aspects. Thus, it is desired to design a universal framework for the adaptive gradient methods on both theoretical analysis and practical algorithms to solve the generic problems.

To fill this gap, in the paper, we propose a faster and universal framework of adaptive gradients, *i.e.*, SUPER-ADAM algorithm, by introducing a universal adaptive matrix. Moreover, we provide a novel convergence analysis framework for the adaptive gradient methods under the nonconvex setting based on the mirror descent algorithm [5, 15]. In summary, our main **contributions** are threefold:

1) We propose a faster and universal framework of adaptive gradients (*i.e.*, SUPER-ADAM) by introducing a universal adaptive matrix that includes most existing adaptive gradients. Moreover, our framework can flexibly integrate the momentum and variance-reduced techniques.

2) We provide a novel convergence analysis framework for the adaptive gradient methods in the nonconvex setting under the milder conditions (Please see Table 1).

3) We apply a momentum-based variance reduced gradient estimator [11, 32] to our algorithm (SUPER-ADAM ($\tau = 1$)), which makes our algorithm reach a faster convergence rate than the classic adaptive methods. Specifically, under smoothness of each component function $f(x; \xi)$, we prove that the SUPER-ADAM ($\tau = 1$) achieves the best known gradient complexity of $\tilde{O}(\epsilon^{-3})$ for finding an $\epsilon$-stationary point of the problem (1), which matches the lower bound for stochastic smooth nonconvex optimization [1]. Under smoothness of the function $f(x)$, we prove that the SUPER-ADAM ($\tau = 0$) achieves a gradient complexity of $\tilde{O}(\epsilon^{-4})$.

## 2 Preliminaries

### 2.1 Notations

$\| \cdot \|$ denotes the $\ell_2$ norm for vectors and spectral norm for matrices, respectively. $I_d$ denotes a $d$-dimensional identity matrix. $\text{diag}(a) \in \mathbb{R}^d$ denotes a diagonal matrix with diagonal entries $a = (a_1, \cdots, a_d)$. For vectors $u$ and $v$, $u^p$ ($p > 0$) denotes element-wise power operation, $u/v$ denotes element-wise division and $\max(u, v)$ denotes element-wise maximum. $\langle u, v \rangle$ denotes the inner product of two vectors $u$ and $v$. For two sequences $\{a_n\}$ and $\{b_n\}$, we write $a_n = O(b_n)$ if there exists a positive constant $C$ such that $a_n \leq Cb_n$, and $\tilde{O}(\cdot)$ hides logarithmic factors. $A \succ 0 (\succeq 0)$ denotes a positive (semi)definite matrix. $\delta_{\min}(A)$ and $\delta_{\max}(A)$ denote the smallest and largest eigenvalues of the matrix $A$, respectively.

### 2.2 Adaptive Gradient Algorithms

In the subsection, we review some existing typical adaptive gradient methods. Recently, many adaptive algorithms have been proposed to solve the problem (1), and achieve good performances. For example, Adagrad [12] is the first adaptive gradient method with adaptive learning rate for each individual dimension, which adopts the following update form:

$$x_{t+1} = x_t - \eta_t g_t / \sqrt{v_t}, \tag{2}$$

where $g_t = \nabla f(x_t; \xi_t)$ and $v_t = \frac{1}{t} \sum_{j=1}^{t} g_j^2$, and $\eta_t = \frac{\eta}{\sqrt{t}}$ with $\eta > 0$ is the step size. In fact, $\eta_t$ only is the basic learning rate that is the same for all coordinates of variable $x_t$, while $\frac{\eta_t}{\sqrt{v_{t,i}}}$ is the effective learning rate for the $i$-th coordinate of $x_t$, which changes across the coordinates.

Adam [22] is one of the most popular exponential moving average variant of Adagrad, which combines the exponential moving average technique with momentum acceleration. Its update form is:

$$m_t = \alpha_1 m_{t-1} + (1 - \alpha_1) \nabla f(x_t; \xi_t), \quad v_t = \alpha_2 v_{t-1} + (1 - \alpha_2)(\nabla f(x_t; \xi_t))^2$$

$$\hat{m}_t = m_t / (1 - \alpha_1^t), \quad \hat{v}_t = v_t / (1 - \alpha_2^t), \quad x_{t+1} = x_t - \eta_t \hat{m}_t / (\sqrt{\hat{v}_t} + \varepsilon), \quad \forall t \geq 1 \tag{3}$$

where $\alpha_1, \alpha_2 \in (0, 1)$ and $\varepsilon > 0$, and $\eta_t = \frac{\eta}{\sqrt{t}}$ with $\eta > 0$. However, Reddi et al. [29] found a divergence issue of the Adam algorithm, and proposed a modified version of Adam (i.e., Amsgrad), which adopts a new step instead of the debiasing step in (3) to ensure the decay of the effective learning rate, defined as

$$\hat{v}_t = \max(\hat{v}_{t-1}, v_t), \quad x_{t+1} = x_t - \eta_t m_t / \sqrt{\hat{v}_t}. \tag{4}$$

**Algorithm 1** SUPER-ADAM Algorithm

1: **Input:** Total iteration $T$, and tuning parameters $\{\mu_t, \alpha_t\}_{t=1}^T, \gamma > 0$ ;
2: **Initialize:** $x_1 \in \mathcal{X}$, sample one point $\xi_1$ and compute $g_1 = \nabla f(x_1; \xi_1)$;
3: **for** $t = 1, 2, \ldots, T$ **do**
4:     Generate an adaptive matrix $H_t \in \mathbb{R}^{d \times d}$;   // Given two examples to update $H_t$:
5:     Case 1: given $\beta \in (0,1)$, $\lambda > 0$ and $v_0 = 0$,
6:     $v_t = \beta v_{t-1} + (1 - \beta)\nabla f(x_t; \xi_t)^2$, $H_t = \text{diag}(\sqrt{v_t} + \lambda)$;
7:     Case 2: given $\beta \in (0,1)$, $\lambda > 0$ and $b_0 = 0$,
8:     $b_t = \beta b_{t-1} + (1 - \beta)\|\nabla f(x_t; \xi_t)\|$, $H_t = (b_t + \lambda)I_d$;
9:     Update $\tilde{x}_{t+1} = \arg\min_{x \in \mathcal{X}} \left\{ \langle g_t, x \rangle + \frac{1}{2\gamma}(x - x_t)^T H_t(x - x_t) \right\}$;
10:    Update $x_{t+1} = (1 - \mu_t)x_t + \mu_t \tilde{x}_{t+1}$;
11:    Sample one point $\xi_{t+1}$, and compute $g_{t+1} = \alpha_{t+1}\nabla f(x_{t+1}; \xi_{t+1}) + (1 - \alpha_{t+1})\big[g_t + \tau\big(\nabla f(x_{t+1}; \xi_{t+1}) - \nabla f(x_t; \xi_{t+1})\big)\big]$, where $\tau \in \{0, 1\}$;
12: **end for**
13: **Output:** (for theoretical) $x_\zeta$ chosen uniformly random from $\{x_t\}_{t=1}^T$; (for practical ) $x_T$.

Due to using the coordinate-wise learning rates, these adaptive gradient methods frequently have worse generalization performance than SGD (with momentum) [35]. To improve the generalization performance of Adam, AdamW [26] uses a decoupled weight decay regularization, defined as

$$m_t = \alpha_1 m_{t-1} + (1 - \alpha_1)\nabla f(x_t; \xi_t), \quad v_t = \alpha_2 v_{t-1} + (1 - \alpha_2)(\nabla f(x_t; \xi_t))^2$$
$$\hat{m}_t = m_t/(1 - \alpha_1^t), \quad \hat{v} = v_t/(1 - \alpha_2^t), \quad x_{t+1} = x_t - \eta_t\big(\alpha \hat{m}_t/(\sqrt{\hat{v}_t} + \varepsilon) + \lambda x_t\big), \quad (5)$$

where $\alpha_1, \alpha_2 \in (0,1)$, $\alpha > 0$, $\lambda > 0$ and $\varepsilon > 0$. More recently, to further improve generalization performance, AdaBelief [40] adopts a stepsize according to 'belief' in the current gradient direction,

$$m_t = \alpha_1 m_{t-1} + (1 - \alpha_1)\nabla f(x_t; \xi_t), \quad v_t = \alpha_2 v_{t-1} + (1 - \alpha_2)(\nabla f(x_t; \xi_t) - m_t)^2 + \varepsilon$$
$$\hat{m}_t = m_t/(1 - \alpha_1^t), \quad \hat{v}_t = v_t/(1 - \alpha_2^t), \quad x_{t+1} = x_t - \eta_t \hat{m}_t/(\sqrt{\hat{v}_t} + \varepsilon), \quad \forall\, t \geq 1 \quad (6)$$

where $\alpha_1, \alpha_2 \in (0, 1)$, and $\eta_t = \frac{\eta}{\sqrt{t}}$ with $\eta > 0$, and $\varepsilon > 0$.

At the same time, to improve generalization performance, recently some effective adaptive gradient methods [34, 23, 11] have been proposed with adopting the global adaptive learning rates instead of coordinate-wise counterparts. For example, AdaGrad-Norm [34] applies a global adaptive learning rate to the following update form, for all $t \geq 1$

$$x_t = x_{t-1} - \eta\nabla f(x_{t-1}; \xi_{t-1})/b_t, \quad b_t^2 = b_{t-1}^2 + \|\nabla f(x_{t-1}; \xi_{t-1})\|^2, \; b_0 > 0, \quad (7)$$

where $\eta > 0$. The adaptive-SGD [23] adopts a global adaptive learning rate, defined as for all $t \geq 1$

$$\eta_t = \frac{k}{\big(\omega + \sum_{i=1}^{t-1} \|\nabla f(x_i; \xi_i)\|^2\big)^{1/2 + \varepsilon}}, \quad x_{t+1} = x_t - \eta_t \nabla f(x_t; \xi_t), \quad (8)$$

where $k > 0$, $\omega > 0$, and $\varepsilon \geq 0$. Subsequently, STORM [11] not only uses a global adaptive learning rate but also adopts the variance-reduced technique in gradient estimator to accelerate algorithm, defined as for all $t \geq 1$

$$\eta_t = \frac{k}{\big(\omega + \sum_{i=1}^t \|\nabla f(x_i; \xi_i)\|^2\big)^{1/3}}, \quad x_{t+1} = x_t - \eta_t g_t,$$
$$g_{t+1} = \nabla f(x_{t+1}; \xi_{t+1}) + (1 - c\eta_t^2)(g_t - \nabla f(x_t; \xi_{t+1})), \quad (9)$$

where $k > 0$, $\omega > 0$ and $c > 0$.

## 3 SUPER-ADAM Algorithm

In the section, we propose a faster and universal framework of adaptive gradients (i.e., SUPER-ADAM) by introducing a universal adaptive matrix that includes most existing adaptive gradient forms. Specifically, our SUPER-ADAM algorithm is summarized in Algorithm 1.

At the step 4 in Algorithm 1, we generate an adaptive matrix $H_t$ based on stochastic gradient information, which can include both coordinate-wise and global learning rates. For example, $H_t$ generated from the **case 1** in Algorithm 1 is similar to the coordinate-wise adaptive learning rate used in Adam [22]. $H_t$ generated from the **case 2** in Algorithm 1 is similar to the global adaptive learning rate used in the AdaGrad-Norm [34] and Adaptive-SGD [23]. Moreover, we can obtain some new adaptive learning rates by generating some specific adaptive matrices. In the **case 3**, based on Barzilai-Borwein technique [2], we design a novel adaptive matrix $H_t$ defined as:

$$b_t = \frac{|\langle \nabla f(x_t; \xi_t) - \nabla f(x_{t-1}; \xi_t), x_t - x_{t-1}\rangle|}{\|x_t - x_{t-1}\|^2}, \quad H_t = (b_t + \lambda)I_d, \tag{10}$$

where $\lambda > 0$. In the **case 4**, as the adaptive learning rate used in [40], we can generate a coordinate-wise-type adaptive matrix $H_t = \text{diag}(\sqrt{v_t} + \lambda)$ and a global-type adaptive matrix $H_t = (b_t + \lambda)I_d$, respectively, defined as: $m_t = \beta_1 m_{t-1} + (1 - \beta_1)\nabla f(x_t; \xi_t)$,

$$v_t = \beta_2 v_{t-1} + (1 - \beta_2)(\nabla f(x_t; \xi_t) - m_t)^2, \quad b_t = \beta_2 b_{t-1} + (1 - \beta_2)\|\nabla f(x_t; \xi_t) - m_t\|, \tag{11}$$

where $\beta_1, \beta_2 \in (0, 1)$ and $\lambda > 0$. In fact, the adaptive matrix $H_t$ can be given in a generic form $H_t = A_t + \lambda I_d$, where the matrix $A_t$ includes the adaptive information that is generated from stochastic gradients with noises, and the tuning parameter $\lambda > 0$ balances these adaptive information with noises.

At the step 9 in Algorithm 1, we use a generalized gradient descent (i.e., mirror descent) iteration [5, 3, 15] to update $x$ based on the adaptive matrix $H_t$, defined as

$$\tilde{x}_{t+1} = \arg\min_{x \in \mathcal{X}} \left\{ \langle g_t, x \rangle + \frac{1}{2\gamma}(x - x_t)^T H_t(x - x_t) \right\} \tag{12}$$

$$= \arg\min_{x \in \mathcal{X}} \left\{ f(x_t) + \langle g_t, x - x_t \rangle + \frac{1}{2\gamma}(x - x_t)^T H_t(x - x_t) \right\}, \tag{13}$$

where $\gamma > 0$ is a constant stepsize. In the above subproblem (13), we can omit the constant terms $f(x_t)$ and $\langle g_t, x_t \rangle$. For the subproblem (13), the first two terms of its objective function is a linear function approximated the function $f(x)$ based on the stochastic gradient $g_t$, and the last term can be seen as a Bregman distance between $x$ and $x_t$ based on the Bregman function $w_t(x) = \frac{1}{2}x^T H_t x$. At the step 10 in Algorithm 1, we use momentum update to obtain a weighted solution $x_{t+1} = (1 - \mu_t)x_t + \mu_t \tilde{x}_{t+1}$, where $\mu_t \in (0, 1]$ ensures $x_{t+1} \in \mathcal{X}$. When $\mathcal{X} = \mathbb{R}^d$, the step 9 is equivalent to $\tilde{x}_{t+1} = x_t - \gamma H_t^{-1} g_t$. Then by the step 10, we have

$$x_{t+1} = (1 - \mu_t)x_t + \mu_t \tilde{x}_{t+1} = x_t - \gamma \mu_t H_t^{-1} g_t. \tag{14}$$

Under this case, $\gamma\mu_t$ is a basic stepsize as $\eta_t$ in the formula (3) of Adam algorithm, and $H_t^{-1}$ is an adaptive stepsize as $\frac{1}{\sqrt{\hat{v}_t}}$ in the formula (3) of Adam algorithm.

At the step 11 of Algorithm 1, we use the stochastic gradient estimator $g_{t+1}$ for all $t \geq 1$:

$$g_{t+1} = \alpha_{t+1}\nabla f(x_{t+1}; \xi_{t+1}) + (1 - \alpha_{t+1})\left[g_t + \tau\left(\nabla f(x_{t+1}; \xi_{t+1}) - \nabla f(x_t; \xi_{t+1})\right)\right], \tag{15}$$

where $\tau \in \{0, 1\}$ and $\alpha_{t+1} \in (0, 1]$ for all $t \geq 1$. When $\tau = 1$, we have $g_{t+1} = \nabla f(x_{t+1}; \xi_{t+1}) + (1 - \alpha_{t+1})\left(g_t - \nabla f(x_t; \xi_{t+1})\right)$ for all $t \geq 1$, which is a momentum-based variance reduced gradient estimator used in STORM [11]. When $\tau = 0$, we have $g_{t+1} = \alpha_{t+1}\nabla f(x_{t+1}; \xi_{t+1}) + (1 - \alpha_{t+1})g_t$ for all $t \geq 1$, which is a basic momentum gradient estimator used in the Adam algorithm [22].

## 4 Theoretical Analysis

In this section, we study the convergence properties of our algorithm (SUPER-ADAM) under some mild conditions. All detailed proofs are in the supplementary materials.

### 4.1 Some Mild Assumptions

**Assumption 1.** *Variance of unbiased stochastic gradient is bounded,* i.e., *there exists a constant $\sigma > 0$ such that for all $x \in \mathcal{X}$, it follows $\mathbb{E}[\nabla f(x; \xi)] = \nabla f(x)$ and $\mathbb{E}\|\nabla f(x; \xi) - \nabla f(x)\|^2 \leq \sigma^2$.*

**Assumption 2.** *The function $f(x)$ is bounded from below in $\mathcal{X}$,* i.e., $f^* = \inf_{x \in \mathcal{X}} f(x)$.

**Assumption 3.** *Assume the adaptive matrix $H_t$ for all $t \geq 1$ satisfies $H_t \succeq \rho I_d \succ 0$, and $\rho > 0$ denotes a lower bound of the smallest eigenvalue of $H_t$ for all $t \geq 1$.*

Assumption 1 is commonly used in stochastic optimization [15, 11]. Assumption 2 ensures the feasibility of the problem (1). In fact, all adaptive algorithms in Table 1 require these mild Assumptions 1 and 2. Assumption 3 guarantees that the adaptive matrices $\{H_t\}_{t\geq 1}$ are positive definite and their smallest eigenvalues have a lower bound $\rho > 0$. From the above adaptive matrices $\{H_t\}_{t\geq 1}$ given in our SUPER-ADAM algorithm, we have $\rho \geq \lambda > 0$. In fact, many existing adaptive algorithms also implicitly use Assumption 3. For example, Zaheer et al. [36] and Zhuang et al. [40] used the following iteration form to update the variable $x$: $x_{t+1} = x_t - \eta_t \frac{m_t}{\sqrt{v_t}+\varepsilon}$ for all $t \geq 0$ and $\varepsilon > 0$, which is equivalent to $x_{t+1} = x_t - \eta_t H_t^{-1} m_t$ with $H_t = \mathrm{diag}(\sqrt{v_t} + \varepsilon)$. Clearly, we have $H_t \succeq \varepsilon I_d \succ 0$. Ward et al. [34] applied a global adaptive learning rate to the update form in (7), which is equivalent to the following form: $x_t = x_{t-1} - \eta H_t^{-1} \nabla f(x_{t-1}; \xi_{t-1})$ with $H_t = b_t I_d$. By the above (7), we have $H_t \succeq \cdots \succeq H_0 = b_0 I_d \succ 0$. Li et al. [23] and Cutkosky et al. [11] applied a global adaptive learning rate to the update forms in (8) and (9), which is equivalent to $x_{t+1} = x_t - H_t^{-1} g_t$, where $H_t = (1/\eta_t) I_d$ and $\eta_t = k / (\omega + \sum_{i=1}^{t} \|\nabla f(x_i; \xi_i)\|^2)^\alpha$ with $k > 0, \omega > 0, \alpha \in (0,1)$. By the above (8) and (9), we have $H_t \succeq \cdots \succeq H_0 = (\omega^\alpha/k) I_d \succ 0$. Reddi et al. [29] and Chen et al. [6] used the condition $\hat{v}_t = \max(\hat{v}_{t-1}, v_t)$, and let $H_t = \mathrm{diag}(\sqrt{\hat{v}_t})$, thus we have $H_t \succeq \cdots \succeq H_1 = \mathrm{diag}(\sqrt{\hat{v}_1}) = \sqrt{1-\alpha_2}\mathrm{diag}(|\nabla f(x_1; \xi_1)|) \succeq 0$. Without loss of generality, choosing an initial point $x_1$ and let $(\nabla f(x_1; \xi_1))_j \neq 0$ for all $j \in [d]$, we have $H_t \succeq \cdots \succeq H_1 \succ 0$. Interestingly, our SUPER-ADAM algorithm includes a class of novel momentum-based quasi-Newton algorithms by generating an approximated Hessian matrix $H_t$. In fact, the quasi-Newton algorithms [33, 16, 38] generally require the bounded approximated Hessian matrices, i.e., $\hat{\kappa} I_d \succeq H_t \succeq \bar{\kappa} I_d \succ 0$ for all $t \geq 1$, where $\hat{\kappa} \geq \bar{\kappa} > 0$. Thus Assumption 3 is reasonable and mild. Due to Assumption 3, our convergence analysis can be easily applied to the stochastic quasi-Newton algorithms.

### 4.2 A Useful Convergence Measure

We provide a useful measure to analyze the convergence of our algorithm, defined as

$$\mathcal{M}_t = \frac{1}{\rho}\|\nabla f(x_t) - g_t\| + \frac{1}{\gamma}\|\tilde{x}_{t+1} - x_t\|. \tag{16}$$

We define a Bregman distance [4, 5, 15] associated with function $w_t(x) = \frac{1}{2}x^T H_t x$ as follows

$$V_t(x, x_t) = w_t(x) - \left[w_t(x_t) + \langle \nabla w_t(x_t), x - x_t \rangle\right] = \frac{1}{2}(x - x_t)^T H_t(x - x_t). \tag{17}$$

Thus, the step 9 of Algorithm 1 is equivalent to the following mirror descent iteration:

$$\tilde{x}_{t+1} = \arg\min_{x \in \mathcal{X}} \left\{ \langle g_t, x \rangle + \frac{1}{\gamma} V_t(x, x_t) \right\}. \tag{18}$$

As in [15], we define a gradient mapping $\mathcal{G}_{\mathcal{X}}(x_t, \nabla f(x_t), \gamma) = \frac{1}{\gamma}(x_t - x_{t+1}^+)$, where

$$x_{t+1}^+ = \arg\min_{x \in \mathcal{X}} \left\{ \langle \nabla f(x_t), x \rangle + \frac{1}{\gamma} V_t(x, x_t) \right\}. \tag{19}$$

Let $\mathcal{G}_{\mathcal{X}}(x_t, g_t, \gamma) = \frac{1}{\gamma}(x_t - \tilde{x}_{t+1})$. According to Proposition 1 in [15], we have $\|\mathcal{G}_{\mathcal{X}}(x_t, g_t, \gamma) - \mathcal{G}_{\mathcal{X}}(x_t, \nabla f(x_t), \gamma)\| \leq \frac{1}{\rho}\|\nabla f(x_t) - g_t\|$. Since $\|\mathcal{G}_{\mathcal{X}}(x_t, \nabla f(x_t), \gamma)\| \leq \|\mathcal{G}_{\mathcal{X}}(x_t, g_t, \gamma)\| + \|\mathcal{G}_{\mathcal{X}}(x_t, g_t, \gamma) - \mathcal{G}_{\mathcal{X}}(x_t, \nabla f(x_t), \gamma)\|$, we have $\|\mathcal{G}_{\mathcal{X}}(x_t, \nabla f(x_t), \gamma)\| \leq \|\mathcal{G}_{\mathcal{X}}(x_t, g_t, \gamma)\| + \frac{1}{\rho}\|\nabla f(x_t) - g_t\| = \frac{1}{\gamma}\|x_t - \tilde{x}_{t+1}\| + \frac{1}{\rho}\|\nabla f(x_t) - g_t\| = \mathcal{M}_t$. When $\mathcal{M}_t \to 0$, we can obtain $\|\mathcal{G}_{\mathcal{X}}(x_t, \nabla f(x_t), \gamma)\| \to 0$, where $x_t$ is a stationary point or local minimum of the problem (1) [15]. Clearly, our measure $\mathbb{E}[\mathcal{M}_t]$ is tighter than the gradient mapping measure $\mathbb{E}\|\mathcal{G}_{\mathcal{X}}(x_t, \nabla f(x_t), \gamma)\|$.

### 4.3 Convergence Analysis of SUPER-ADAM ($\tau = 1$)

In this subsection, we provide the convergence analysis of our SUPER-ADAM ($\tau = 1$) algorithm using the momentum-based variance reduced gradient estimator [11, 32].

**Assumption 4.** *Each component function $f(x; \xi)$ is $L$-smooth for all $\xi \in \mathcal{D}$, i.e.,*

$$\|\nabla f(x; \xi) - \nabla f(y; \xi)\| \leq L\|x - y\|, \ \forall x, y \in \mathcal{X}.$$

Assumption 4 is widely used in the variance-reduced algorithms [13, 11]. According to Assumption 4, we have $\|\nabla f(x) - \nabla f(y)\| = \|\mathbb{E}[\nabla f(x; \xi) - \nabla f(y; \xi)]\| \leq \mathbb{E}\|\nabla f(x; \xi) - \nabla f(y; \xi)\| \leq L\|x - y\|$ for all $x, y \in \mathcal{X}$. Thus the function $f(x)$ also is $L$-smooth.

**Theorem 1.** *In Algorithm 1, under the Assumptions (1,2,3,4), when $\mathcal{X} \subset \mathbb{R}^d$, and given $\tau = 1$, $\mu_t = \frac{k}{(m+t)^{1/3}}$ and $\alpha_{t+1} = c\mu_t^2$ for all $t \geq 0$, $0 < \gamma \leq \frac{\rho m^{1/3}}{4kL}$, $\frac{1}{k^3} + \frac{10L^2\gamma^2}{\rho^2} \leq c \leq \frac{m^{2/3}}{k^2}$, $m \geq \max\left(\frac{3}{2}, k^3, \frac{8^{3/2}}{(3k)^{3/2}}\right)$ and $k > 0$, we have*

$$\frac{1}{T}\sum_{t=1}^{T}\mathbb{E}\|\mathcal{G}_{\mathcal{X}}(x_t, \nabla f(x_t), \gamma)\| \leq \frac{1}{T}\sum_{t=1}^{T}\mathbb{E}[\mathcal{M}_t] \leq \frac{2\sqrt{2G}m^{1/6}}{T^{1/2}} + \frac{2\sqrt{2G}}{T^{1/3}}, \qquad (20)$$

*where $G = \frac{f(x_1) - f^*}{k\rho\gamma} + \frac{m^{1/3}\sigma^2}{8k^2L^2\gamma^2} + \frac{k^2c^2\sigma^2}{4L^2\gamma^2}\ln(m+T)$.*

**Remark 1.** *Without loss of generality, let $\rho = O(1)$, $k = O(1)$, $m = O(1)$, and $\gamma = O(1)$, we have $c = O(1)$ and $G = O(c^2\sigma^2\ln(m+T)) = \tilde{O}(1)$. Thus, our algorithm has a convergence rate of $\tilde{O}\left(\frac{1}{T^{1/3}}\right)$. Let $\frac{1}{T^{1/3}} \leq \epsilon$, we have $T \geq \epsilon^{-3}$. Since our algorithm only requires to compute two stochastic gradients at each iteration (e.g., only need to compute stochastic gradients $\nabla f(x_{t+1}; \xi_{t+1})$ and $\nabla f(x_t; \xi_{t+1})$ to estimate $g_{t+1}$), and needs $T$ iterations. Thus, our SUPER-ADAM ($\tau = 1$) has a gradient complexity of $2 \cdot T = \tilde{O}(\epsilon^{-3})$ for finding an $\epsilon$-stationary point.*

**Corollary 1.** *In Algorithm 1, under the above Assumptions (1,2,3,4), when $\mathcal{X} = \mathbb{R}^d$, and given $\tau = 1$, $\mu_t = \frac{k}{(m+t)^{1/3}}$ and $\alpha_{t+1} = c\mu_t^2$ for all $t \geq 0$, $\gamma = \frac{\rho m^{1/3}}{\nu kL}$ ($\nu \geq 4$), $\frac{1}{k^3} + \frac{10L^2\gamma^2}{\rho^2} \leq c \leq \frac{m^{2/3}}{k^2}$, $m \geq \max\left(\frac{3}{2}, k^3, \frac{8^{3/2}}{(3k)^{3/2}}\right)$ and $k > 0$, we have*

$$\frac{1}{T}\sum_{t=1}^{T}\mathbb{E}\|\nabla f(x_t)\| \leq \frac{\max_{1\leq t\leq T}\|H_t\|}{\rho}\left(\frac{2\sqrt{2G'}}{T^{1/2}} + \frac{2\sqrt{2G'}}{m^{1/6}T^{1/3}}\right), \qquad (21)$$

*where $G' = \nu L(f(x_1) - f^*) + \frac{\nu^2\sigma^2}{8} + \frac{\nu^2k^4c^2\sigma^2}{4m^{1/3}}\ln(m+T)$.*

**Remark 2.** *Under the same conditions in Theorem 1, based on the metric $\mathbb{E}\|\nabla f(x)\|$, our SUPER-ADAM ($\tau = 1$) still has a gradient complexity of $\tilde{O}(\epsilon^{-3})$. Interestingly, the right of the above inequality (21) includes a term $\frac{\max_{1\leq t\leq T}\|H_t\|}{\rho}$ that can be seen as an upper bound of the condition number of adaptive matrices $\{H_t\}_{t=1}^{T}$. When using $H_t$ given in the above **case 1**, we have $\frac{\max_{1\leq t\leq T}\|H_t\|}{\rho} \leq \frac{G_1+\lambda}{\lambda}$ as in the existing adaptive gradient methods assuming the bounded stochastic gradient $\|\nabla f(x; \xi)\|_\infty \leq G_1$; When using $H_t$ given in the above **case 2**, we have $\frac{\max_{1\leq t\leq T}\|H_t\|}{\rho} \leq \frac{G_2+\sigma+\lambda}{\lambda}$ as in the existing adaptive gradient methods assuming the bounded full gradient $\|\nabla f(x)\| \leq G_2$; When using $H_t$ given in the above **case 3**, we have $\frac{\max_{1\leq t\leq T}\|H_t\|}{\rho} \leq \frac{L+\lambda}{\lambda}$. When using $H_t$ given in the above **case 4**, we have $\frac{\max_{1\leq t\leq T}\|H_t\|}{\rho} \leq \frac{2G_1+\lambda}{\lambda}$ or $\frac{\max_{1\leq t\leq T}\|H_t\|}{\rho} \leq \frac{2(G_2+\sigma)+\lambda}{\lambda}$. **Note that** we only study the gradient (sample) complexity of our algorithm in **the worst case** without considering some specific structures such as the sparsity of stochastic gradient. Since the adaptive matrix $H_t$ can be given $H_t = A_t + \lambda I_d$, we have $\frac{\max_{1\leq t\leq T}\|H_t\|}{\rho} = \frac{\max_{1\leq t\leq T}\delta_{\max}(A_t)+\lambda}{\min_{1\leq t\leq T}\delta_{\min}(A_t)+\lambda}$. Here we only can choose a proper tuning parameter $\lambda$ to balance adaptive information with noises in $A_t$. To reduce $\frac{\max_{1\leq t\leq T}\|H_t\|}{\rho}$, we can not increase $\lambda$, but should design the matrix $A_t$ with a small condition number by some techniques, e.g., clipping [27].*

### 4.4 Convergence Analysis of SUPER-ADAM ($\tau = 0$)

In this subsection, we provide the convergence analysis of our SUPER-ADAM ($\tau = 0$) algorithm using the basic momentum stochastic gradient estimator [22].

**Assumption 5.** *The function $f(x) = \mathbb{E}_\xi[f(x; \xi)]$ is $L$-smooth, i.e.,*

$$\|\nabla f(x) - \nabla f(y)\| \leq L\|x - y\|, \ \forall x, y \in \mathcal{X}.$$

Assumption 5 is widely used in adaptive algorithms [36, 8, 40], which is milder than Assumption 4.

**Theorem 2.** *In Algorithm 1, under the Assumptions (1,2,3,5), when $\mathcal{X} \subset \mathbb{R}^d$, and given $\tau = 0$, $\mu_t = \frac{k}{(m+t)^{1/2}}$, $\alpha_{t+1} = c\mu_t$ for all $t \geq 0$, $k > 0$, $0 < \gamma \leq \frac{\rho m^{1/2}}{8Lk}$, $\frac{8L\gamma}{\rho} \leq c \leq \frac{m^{1/2}}{k}$, and $m \geq k^2$, we have*

$$\frac{1}{T}\sum_{t=1}^{T}\mathbb{E}\|\mathcal{G}_{\mathcal{X}}(x_t, \nabla f(x_t), \gamma)\| \leq \frac{1}{T}\sum_{t=1}^{T}\mathbb{E}[\mathcal{M}_t] \leq \frac{2\sqrt{2M}m^{1/4}}{T^{1/2}} + \frac{2\sqrt{2M}}{T^{1/4}},$$

where $M = \frac{f(x_1)-f^*}{\rho\gamma k} + \frac{2\sigma^2}{\rho\gamma kL} + \frac{2m\sigma^2}{\rho\gamma kL}\ln(m+T)$.

**Remark 3.** *Without loss of generality, let $\rho = O(1)$, $k = O(1)$, $m = O(1)$ and $\gamma = O(1)$, we have $M = O\big(\sigma^2\ln(m+T)\big) = \tilde{O}(1)$. Thus, our algorithm has convergence rate of $\tilde{O}\big(\frac{1}{T^{1/4}}\big)$. Considering $\frac{1}{T^{1/4}} \leq \epsilon$, we have $T \geq \epsilon^{-4}$. Since our algorithm requires to compute one stochastic gradient at each iteration, and needs $T$ iterations. Thus, our SUPER-ADAM ($\tau = 0$) has a gradient complexity of $1 \cdot T = \tilde{O}(\epsilon^{-4})$ for finding an $\epsilon$-stationary point.*

**Corollary 2.** *In Algorithm 1, under the above Assumptions (1,2,3,5), when $\mathcal{X} = \mathbb{R}^d$, and given $\tau = 0$, $\mu_t = \frac{k}{(m+t)^{1/2}}$, $\alpha_{t+1} = c\mu_t$ for all $t \geq 0$, $k > 0$, $\gamma = \frac{\rho m^{1/2}}{\nu Lk}$ ($\nu \geq 8$), $\frac{8L\gamma}{\rho} \leq c \leq \frac{m^{1/2}}{k}$, and $m \geq k^2$, we have*

$$\frac{1}{T}\sum_{t=1}^{T}\mathbb{E}\|\nabla f(x_t)\| \leq \frac{\max_{1\leq t\leq T}\|H_t\|}{\rho}\left(\frac{2\sqrt{2M'}}{T^{1/2}} + \frac{2\sqrt{2M'}}{m^{1/4}T^{1/4}}\right),$$

*where $M' = \nu L(f(x_1) - f^*) + 2\nu\sigma^2 + 2\nu m\sigma^2\ln(m+T)$.*

**Remark 4.** *Under the same conditions in Theorem 2, based on the metric $\mathbb{E}\|\nabla f(x_t)\|$, our SUPER-ADAM ($\tau = 0$) still has a gradient complexity of $\tilde{O}(\epsilon^{-4})$ for finding an $\epsilon$-stationary point.*

## 5 Differences between Our Algorithm and Related Algorithms

In this section, we show some significance differences between our algorithm and some related algorithms, i.e., STORM algorithm [11] and Adam-type algorithms [22, 29, 40]. Although our SUPER-ADAM ($\tau = 1$) algorithm uses the same stochastic gradient estimator used in the STORM, there exist some significant differences:

1) Our algorithm focuses on both constrained and unconstrained optimizations, but STORM only focuses on unconstrained optimization.

2) In our algorithm, we introduce a weighted solution $x_{t+1}$ at the step 10 by using momentum update. Under this case, our algorithm can easily incorporate various adaptive learning rates and variance reduced techniques. Specifically, we can flexibly use various adaptive learning rates and different stochastic gradient estimators $g_t$ at the step 9 of our algorithm. In fact, this is one of important novelties of our paper. However, the STORM only uses a simple gradient descent iteration with a specific monotonically decreasing adaptive learning rate.

Similarly, although our SUPER-ADAM ($\tau = 0$) algorithm uses the same stochastic gradient estimator used in these Adam-type algorithms, there exist some significant differences besides using different adaptive learning rates. These Adam-type algorithms use a decreasing learning rate $\eta_t = \frac{\eta}{\sqrt{t}}$ (Please see the above (3), (4) and (6)), while our algorithm only uses a constant learning rate $\gamma$ besides an adaptive learning rate. Moreover, our algorithm introduces a weighted solution $x_{t+1}$ at the step 10 with a decreasing parameter $\mu_t = \frac{k}{\sqrt{m+t}}$ (Please see Theorem 2) and uses a decreasing parameter $\alpha_{t+1} = c\mu_t$ in the gradient estimator, while these Adam-type algorithms only use a constant parameter $\alpha_1 \in (0, 1)$ in their gradient estimators. Under this case, our algorithm uses these decreasing parameters $\mu_t$ and $\alpha_{t+1}$ to control the noises in our gradient estimator, so our algorithm does not require some additional assumptions such as the bounded (stochastic) gradient assumption in our convergence analysis for the constrained optimization. For example, when $\tau = 0$, our gradient estimator is $g_{t+1} = \alpha_{t+1}\nabla f(x_{t+1}; \xi_{t+1}) + (1 - \alpha_{t+1})g_t$. Intuitively, with growing $t$, $\alpha_{t+1} = \frac{ck}{\sqrt{m+t}}$ will become small, so the new noises added in our gradient estimator $g_{t+1}$ will also become less.

## 6 Numerical Experiments

In this section, we conduct some experiments to empirically evaluate our SUPER-ADAM algorithm on two deep learning tasks as in [25]: **image classification** on CIFAR-10, CIFAR-100 and Image-Net datasets and **language modeling** on Wiki-Text2 dataset. In the experiments, we compare our SUPER-ADAM algorithm against several state-of-the-art adaptive gradient algorithms, including: (1) SGD, (2) Adam [22], (3) Amsgrad [29], (4) AdaGrad-Norm [23], (5) Adam$^+$ [25], (6) STORM [11] and (7) AdaBelief [40]. For our SUPER-ADAM algorithm, we consider $\tau = 1$ and $\tau = 0$, respectively. Without loss of generality, in the following experiments, we only use the **case 1** in Algorithm 1 to generate adaptive matrix $H_t$ and let $\lambda = 0.0005$. All experiments are run over a machine with Intel Xeon E5-2683 CPU and 4 Nvidia Tesla P40 GPUs.

## 6.1 Image Classification Task

In the experiment, we conduct image classification task on CIFAR-10, CIFAR-100 and Image-Net datasets. We perform training over ResNet-18 [20] and VGG-19 [30] on CIFAR-10 and CIFAR-100 datasets, respectively. For all the optimizers, we set the batch size as 128 and trains for 200 epochs. For the learning rates and other hyper-parameters, we do grid search and report the best one for each optimizer. In Adam, Amsgrad and AdaBelief algorithms, we set the learning rate as 0.001. In AdaGrad-Norm, the best learning rate is 17 for CIFAR-10 and 10 for CIFAR-100, respectively. In Adam$^+$, we use the recommended tuning parameters in [25]. In STORM, the best result is obtained when $w = 6$, $k = 10$ and $c = 100$ for CIFAR-10, while $w = 3$, $k = 10$ and $c = 100$ for CIFAR-100. For our SUPER-ADAM algorithm, in both CIFAR-10 and CIFAR-100 datasets, we set $k = 1$, $m = 100$, $c = 40$, $\gamma = 0.001$ when $\tau = 1$, and $k = 1$, $m = 100$, $c = 20$, $\gamma = 0.001$ when $\tau = 0$. Note that although $c > \frac{m^{2/3}}{k^2}$ ($c > \frac{m^{1/2}}{k}$) in our algorithm, we set $\alpha_t = \min(\alpha_t, 0.9)$ at the first several iterations. In our algorithm, $\mu_t = \frac{k}{(m+t)^{1/3}}$ ($\mu_t = \frac{k}{(m+t)^{1/2}}$) decreases as the number of iteration $t$ increases, so $\alpha_{t+1} = c\mu_t^2$ ($\alpha_{t+1} = c\mu_t$) will be less than 1 after the first several iterations.

We train a ResNet-34 [20] on ImageNet dataset. For all the optimizers, we set the batch size as 256 and trains for 60 epochs. In Adam, Amsgrad and AdaBelief, we set learning rate as 0.001. In AdaGrad-Norm, the best learning rate is 30. In Adam$^+$, we set learning rate as 0.1. In STORM, the best result is obtained when $k = 5$, $w = 100$ and $c = 10$. For our algorithm, we set $k = 1$, $m = 100$, $c = 40$, $\gamma = 0.01$ when $\tau = 1$, and $k = 1$, $m = 100$, $c = 4$, $\gamma = 0.04$ when $\tau = 0$.

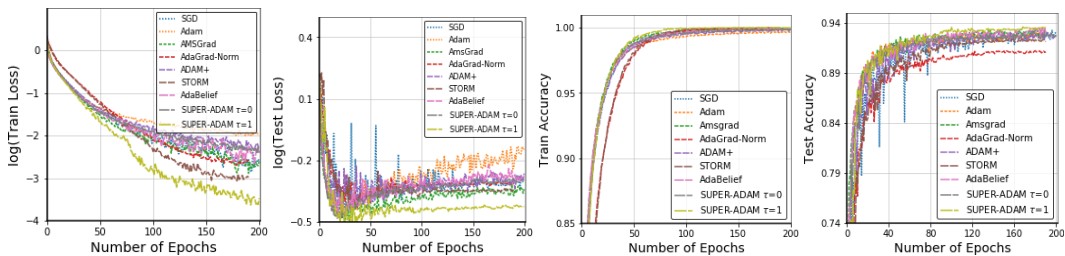

Figure 1: Experimental Results of CIFAR-10 by Different Optimizers over ResNet-18.

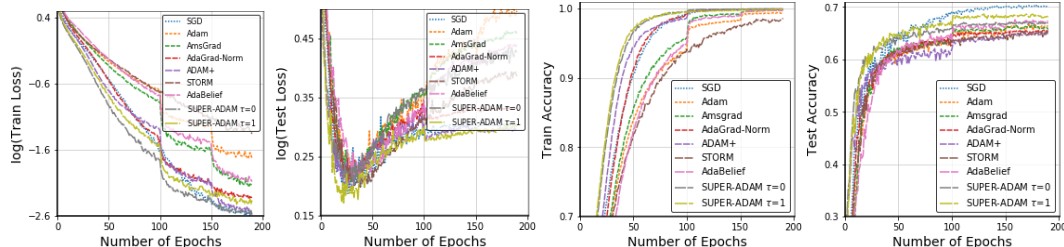

Figure 2: Experimental Results of CIFAR-100 by Different Optimizers over VGG-19.

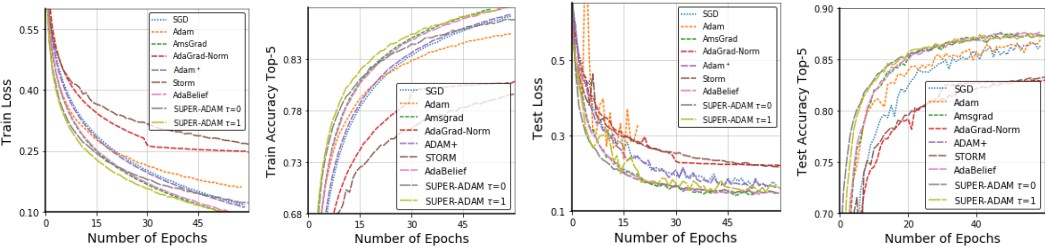

Figure 3: Experimental Results of Image-Net by Different Optimizers over ResNet-34.

Figures 1 and 2 show that both train and test errors and accuracy results of the CIFAR-10 and CIFAR-100 datasets, respectively. Our SUPER-ADAM algorithm consistently outperforms the other optimizers with a great margin, especially when we set $\tau = 1$. When we set $\tau = 0$, our SUPER-ADAM algorithm gets the comparable performances with Adam/AmsGrad. Figure 3 demonstrates

the results of ImageNet by different optimizers over ResNet-34, which shows that our algorithm outperforms the other optimizers, especially set $\tau = 1$ in our algorithm. Figure 4 shows that both the condition number of $H_t$ and the $\ell_2$ norm of full gradient (i.e.,$\|\nabla f(x_t)\|$) decrease as the number of iteration increases. From these results, we find that since the condition number of $H_t$ decreases as the number of iteration increases, so it must has an upper bound. Thus, these experimental results further demonstrate that the above convergence results in Corollaries 1 and 2 are reasonable.

## 6.2  Language Modeling Task

In the experiment, we conduct language modeling task on the Wiki-Text2 dataset. Specifically, we train a 2-layer LSTM [21] and a 2-layer Transformer over the WiKi-Text2 dataset. For the LSTM, we use 650 dimensional word embeddings and 650 hidden units per-layer. Due to space limitation, we provide the experimental results for the transformer in the supplementary materials. In the experiment, we set the batch size as 20 and trains for 40 epochs with dropout rate 0.5. We also clip the gradients by norm 0.25 in case of the exploding gradient in LSTM. We also decrease the learning by 4 whenever the validation error increases. For the learning rate, we also do grid search and report the best one for each optimizer. In Adam and Amsgrad algorithms, we set the learning rate as 0.001 in

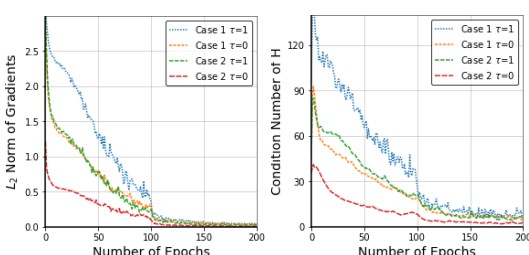

Figure 4: The condition number of $H_t$ and the $\ell_2$ norm of full gradient (i.e.,$\|\nabla f(x_t)\|$) for CIFAR-10 by our SUPER-ADAM algorithm $\tau = 1$ and $\tau = 0$, respectively. Cases 1 and 2 are two choices of adaptive matrix $H_t$ given in Algorithm 1.

LSTM In AdaGrad-Norm algorithm, the best learning rate is 40. In Adam$^+$ algorithm, we use the learning rate 20. In AdaBelief algorithm, we set the learing rate 0.1. In STORM algorithm, we set $w = 50$, $k = 10$ and $c = 100$. In our SUPER-ADAM algorithm, we set $k = 1$, $m = 100$, $c = 40$, $\gamma = 0.001$ when $\tau = 1$, while $k = 1$, $m = 100$, $c = 20$, $\gamma = 0.01$ when $\tau = 0$.

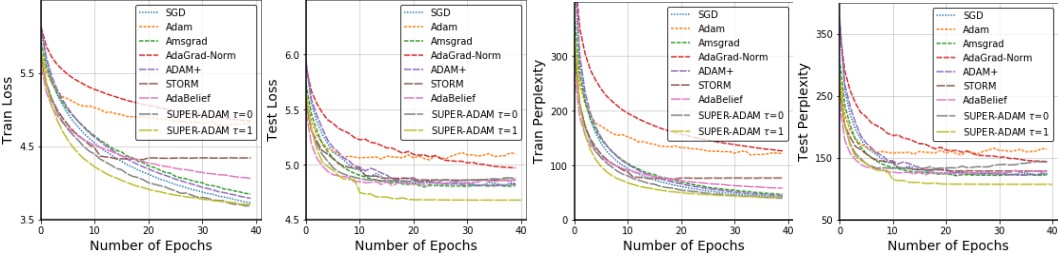

Figure 5: Experimental Results of WikiText-2 by Different Optimizers over LSTM.

Figure 5 shows that both train and test perplexities (losses) for different optimizers. When $\tau = 1$, our SUPER-ADAM algorithm outperforms all the other optimizers. When $\tau = 0$, our SUPER-ADAM optimizer gets a comparable performance with the other Adam-type optimizers.

## 7  Conclusions

In the paper, we proposed a novel faster and universal adaptive gradient framework (i.e., SUPER-ADAM) by introducing a universal adaptive matrix including most existing adaptive gradient forms. In particular, our algorithm can flexibly work with the momentum and variance reduced techniques. Moreover, we provided a novel convergence analysis framework for the adaptive gradient methods under the nonconvex setting. Experimental studies were conducted on both image classification and language modeling tasks, and all empirical results verify the superior performances of our algorithm.

## Acknowledgments and Disclosure of Funding

This work was partially supported by NSF IIS 1845666, 1852606, 1838627, 1837956, 1956002, OIA 2040588.

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
