## A  Proofs of Convergence Analysis

In this section, we detail the convergence analysis of our algorithm. We first provide some useful lemmas.

Given a $\rho$-strongly convex function $w(x)$, we define a prox-function (i.e., Bregman distance) [4, 5] associated with $w(x)$ as follows:

$$V(y, x) = w(y) - \big[ w(x) + \langle \nabla w(x), y - x \rangle \big]. \tag{22}$$

Then we define a generalized projection problem as in [15]:

$$x^* = \arg\min_{y \in \mathcal{X}} \left\{ \langle y, g \rangle + \frac{1}{\gamma} V(y, x) + h(y) \right\}, \tag{23}$$

where $\mathcal{X} \subseteq \mathbb{R}^d$, $g \in \mathbb{R}^d$ and $\gamma > 0$. Here $h(x)$ is a convex and possibly nonsmooth function. At the same time, we define a generalized gradient (i.e., gradient mapping) as follows:

$$\mathcal{G}_{\mathcal{X}}(x, g, \gamma) = \frac{1}{\gamma}(x - x^*). \tag{24}$$

**Lemma 1.** *(Lemma 1 in [15]) Let $x^*$ be given in (23). Then, for any $x \in \mathcal{X}$, $g \in \mathbb{R}^d$ and $\gamma > 0$, we have*

$$\langle g, \mathcal{G}_{\mathcal{X}}(x, g, \gamma) \rangle \geq \rho \| \mathcal{G}_{\mathcal{X}}(x, g, \gamma) \|^2 + \frac{1}{\gamma} \big[ h(x^*) - h(x) \big], \tag{25}$$

*where $\rho > 0$ depends on $\rho$-strongly convex function $w(x)$.*

When $h(x) = 0$, in the above Lemma 1, we have $\langle g, \mathcal{G}_{\mathcal{X}}(x, g, \gamma) \rangle \geq \rho \| \mathcal{G}_{\mathcal{X}}(x, g, \gamma) \|^2$.

**Lemma 2.** *(Proposition 1 in [15]) Let $x_1^*$ and $x_2^*$ be given in (23) with $g$ replaced by $g_1$ and $g_2$ respectively. Further let $\mathcal{G}_{\mathcal{X}}(x, g_1, \gamma)$ and $\mathcal{G}_{\mathcal{X}}(x, g_2, \gamma)$ be defined in (24) with $x^*$ replaced by $x_1^*$ and $x_2^*$ respectively. Then we have*

$$\| \mathcal{G}_{\mathcal{X}}(x, g_1, \gamma) - \mathcal{G}_{\mathcal{X}}(x, g_2, \gamma) \| \leq \frac{1}{\rho} \| g_1 - g_2 \|. \tag{26}$$

**Lemma 3.** *Suppose that the sequence $\{x_t\}_{t=1}^T$ be generated from Algorithm 1. Let $0 < \mu_t \leq 1$ and $0 < \gamma \leq \frac{\rho}{2L\mu_t}$, then we have*

$$f(x_{t+1}) \leq f(x_t) + \frac{\mu_t \gamma}{\rho} \| \nabla f(x_t) - g_t \|^2 - \frac{\rho \mu_t}{2\gamma} \| \tilde{x}_{t+1} - x_t \|^2. \tag{27}$$

*Proof.* According to Assumption 4 or 5, i.e., the function $f(x)$ is $L$-smooth, we have

$$
\begin{aligned}
f(x_{t+1}) &\leq f(x_t) + \langle \nabla f(x_t), x_{t+1} - x_t \rangle + \frac{L}{2} \| x_{t+1} - x_t \|^2 \\
&= f(x_t) + \langle \nabla f(x_t), \mu_t (\tilde{x}_{t+1} - x_t) \rangle + \frac{L}{2} \| \mu_t (\tilde{x}_{t+1} - x_t) \|^2 \\
&= f(x_t) + \mu_t \underbrace{\langle g_t, \tilde{x}_{t+1} - x_t \rangle}_{=T_1} + \mu_t \underbrace{\langle \nabla f(x_t) - g_t, \tilde{x}_{t+1} - x_t \rangle}_{=T_2} + \frac{L\mu_t^2}{2} \| \tilde{x}_{t+1} - x_t \|^2,
\end{aligned} \tag{28}
$$

where the second equality is due to $x_{t+1} = x_t + \mu_t (\tilde{x}_{t+1} - x_t)$.

According to Assumption 3, i.e., $H_t \succ \rho I_d$ for any $t \geq 1$, the function $w_t(x) = \frac{1}{2} x^T H_t x$ is $\rho$-strongly convex, then we have a prox-function associated with $w_t(x)$ as in [15], defined as

$$V_t(x, x_t) = w_t(x) - \big[ w_t(x_t) + \langle \nabla w_t(x_t), x - x_t \rangle \big] = \frac{1}{2} (x - x_t)^T H_t (x - x_t). \tag{29}$$

According to the above Lemma 1, at step 9 in Algorithm 1, i.e., $\tilde{x}_{t+1} = \arg\min_{x \in \mathcal{X}} \left\{ \langle g_t, x \rangle + \frac{1}{2\gamma} (x - x_t)^T H_t (x - x_t) \right\}$, we have

$$\langle g_t, \frac{1}{\gamma}(x_t - \tilde{x}_{t+1}) \rangle \geq \rho \| \frac{1}{\gamma}(x_t - \tilde{x}_{t+1}) \|^2. \tag{30}$$

Then we obtain

$$T_1 = \langle g_t, \tilde{x}_{t+1} - x_t \rangle \leq -\frac{\rho}{\gamma} \| \tilde{x}_{t+1} - x_t \|^2. \tag{31}$$

Next, consider the bound of the term $T_2$, we have

$$
\begin{aligned}
T_2 &= \langle \nabla f(x_t) - g_t, \tilde{x}_{t+1} - x_t \rangle \\
&\leq \|\nabla f(x_t) - g_t\| \cdot \|\tilde{x}_{t+1} - x_t\| \\
&\leq \frac{\gamma}{\rho} \|\nabla f(x_t) - g_t\|^2 + \frac{\rho}{4\gamma} \|\tilde{x}_{t+1} - x_t\|^2,
\end{aligned}
\tag{32}
$$

where the first inequality is due to the Cauchy-Schwarz inequality and the last is due to Young's inequality. By combining the above inequalities (28), (31) with (32), we obtain

$$
\begin{aligned}
f(x_{t+1}) &\leq f(x_t) + \mu_t \langle \nabla f(x_t) - g_t, \tilde{x}_{t+1} - x_t \rangle + \mu_t \langle g_t, \tilde{x}_{t+1} - x_t \rangle + \frac{L\mu_t^2}{2} \|\tilde{x}_{t+1} - x_t\|^2 \\
&\leq f(x_t) + \frac{\mu_t \gamma}{\rho} \|\nabla f(x_t) - g_t\|^2 + \frac{\rho \mu_t}{4\gamma} \|\tilde{x}_{t+1} - x_t\|^2 - \frac{\rho \mu_t}{\gamma} \|\tilde{x}_{t+1} - x_t\|^2 + \frac{L\mu_t^2}{2} \|\tilde{x}_{t+1} - x_t\|^2 \\
&= f(x_t) + \frac{\mu_t \gamma}{\rho} \|\nabla f(x_t) - g_t\|^2 - \frac{\rho \mu_t}{2\gamma} \|\tilde{x}_{t+1} - x_t\|^2 - \left(\frac{\rho \mu_t}{4\gamma} - \frac{L\mu_t^2}{2}\right) \|\tilde{x}_{t+1} - x_t\|^2 \\
&\leq f(x_t) + \frac{\mu_t \gamma}{\rho} \|\nabla f(x_t) - g_t\|^2 - \frac{\rho \mu_t}{2\gamma} \|\tilde{x}_{t+1} - x_t\|^2,
\end{aligned}
\tag{33}
$$

where the last inequality is due to $0 < \gamma \leq \frac{\rho}{2L\mu_t}$.

$\square$

## A.1  Convergence Analysis of SUPER-ADAM ($\tau = 1$)

In this subsection, we provide the convergence analysis of our SUPER-ADAM ($\tau = 1$) algorithm.

**Lemma 4.** *In Algorithm 1, given $\tau = 1$ and $0 < \alpha_{t+1} \leq 1$ for all $t \geq 0$, we have*

$$
\mathbb{E}\|\nabla f(x_{t+1}) - g_{t+1}\|^2 \leq (1 - \alpha_{t+1})^2 \mathbb{E}\|\nabla f(x_t) - g_t\|^2 + 2(1 - \alpha_{t+1})^2 L^2 \mu_t^2 \mathbb{E}\|\tilde{x}_{t+1} - x_t\|^2 \\
+ 2\alpha_{t+1}^2 \sigma^2.
\tag{34}
$$

*Proof.* This proof mainly follows the proof of Lemma 2 in [11]. By the definition of $g_{t+1}$ in Algorithm 1 with $\tau = 1$, we have $g_{t+1} = \nabla f(x_{t+1}; \xi_{t+1}) + (1 - \alpha_{t+1})(g_t - \nabla f(x_t; \xi_{t+1}))$. Then we have

$$
\begin{aligned}
&\mathbb{E}\|\nabla f(x_{t+1}) - g_{t+1}\|^2 &\tag{35} \\
&= \mathbb{E}\|\nabla f(x_t) - g_t + \nabla f(x_{t+1}) - \nabla f(x_t) - (g_{t+1} - g_t)\|^2 \\
&= \mathbb{E}\|\nabla f(x_t) - g_t + \nabla f(x_{t+1}) - \nabla f(x_t) + \alpha_{t+1} g_t - \alpha_{t+1} f(x_{t+1}; \xi_{t+1}) \\
&\quad - (1 - \alpha_{t+1})\left(\nabla f(x_{t+1}; \xi_{t+1}) - \nabla f(x_t; \xi_{t+1})\right)\|^2 \\
&= \mathbb{E}\|(1 - \alpha_{t+1})(\nabla f(x_t) - g_t) + \alpha_{t+1}(\nabla f(x_{t+1}) - \nabla f(x_{t+1}; \xi_{t+1})) \\
&\quad - (1 - \alpha_{t+1})\left(\nabla f(x_{t+1}; \xi_{t+1}) - \nabla f(x_t; \xi_{t+1}) - (\nabla f(x_{t+1}) - \nabla f(x_t))\right)\|^2 \\
&= (1 - \alpha_{t+1})^2 \mathbb{E}\|\nabla f(x_t) - g_t\|^2 + \mathbb{E}\big[\|\alpha_{t+1}(\nabla f(x_{t+1}) - \nabla f(x_{t+1}; \xi_{t+1})) \\
&\quad - (1 - \alpha_{t+1})\left(\nabla f(x_{t+1}; \xi_{t+1}) - \nabla f(x_t; \xi_{t+1}) - (\nabla f(x_{t+1}) - \nabla f(x_t))\right)\|^2\big] \\
&\leq (1 - \alpha_{t+1})^2 \mathbb{E}\|\nabla f(x_t) - g_t\|^2 + 2\alpha_{t+1}^2 \mathbb{E}\|\nabla f(x_{t+1}) - \nabla f(x_{t+1}; \xi_{t+1})\|^2 \\
&\quad + 2(1 - \alpha_{t+1})^2 \mathbb{E}\|\nabla f(x_{t+1}; \xi_{t+1}) - \nabla f(x_t; \xi_{t+1}) - (\nabla f(x_{t+1}) - \nabla f(x_t))\|^2 \\
&\leq (1 - \alpha_{t+1})^2 \mathbb{E}\|\nabla f(x_t) - g_t\|^2 + 2\alpha_{t+1}^2 \sigma^2 + 2(1 - \alpha_{t+1})^2 \mathbb{E}\|\nabla f(x_{t+1}; \xi_{t+1}) - \nabla f(x_t; \xi_{t+1})\|^2 \\
&\leq (1 - \alpha_{t+1})^2 \mathbb{E}\|\nabla f(x_t) - g_t\|^2 + 2\alpha_{t+1}^2 \sigma^2 + 2(1 - \alpha_{t+1})^2 L^2 \mu_t^2 \mathbb{E}\|\tilde{x}_{t+1} - x_t\|^2, &\tag{36}
\end{aligned}
$$

where the fourth equality is due to $\mathbb{E}_{\xi_{t+1}}[\nabla f(x_{t+1}; \xi_{t+1})] = \nabla f(x_{t+1})$ and $\mathbb{E}_{\xi_{t+1}}[\nabla f(x_{t+1}; \xi_{t+1}) - \nabla f(x_t; \xi_{t+1})] = \nabla f(x_{t+1}) - \nabla f(x_t)$, i.e., $\mathbb{E}_{\xi_{t+1}}\big[\alpha_{t+1}(\nabla f(x_{t+1}) - \nabla f(x_{t+1}; \xi_{t+1})) - (1 - \alpha_{t+1})\big(\nabla f(x_{t+1}; \xi_{t+1}) - \nabla f(x_t; \xi_{t+1}) - (\nabla f(x_{t+1}) - \nabla f(x_t))\big)\big] = 0$; the second inequality holds by Assumption 1, and the last inequality is due to Assumption 4 and $x_{t+1} = x_t + \mu_t(\tilde{x}_{t+1} - x_t)$. $\square$

**Theorem 3.** *(Restatement of Theorem 1) In Algorithm 1, under the above Assumptions (1,2,3,4), when $\mathcal{X} \subset \mathbb{R}^d$, and given $\tau = 1$, $\mu_t = \frac{k}{(m+t)^{1/3}}$ and $\alpha_{t+1} = c\mu_t^2$ for all $t \geq 0$, $0 < \gamma \leq \frac{\rho m^{1/3}}{4kL}$, $\frac{1}{k^3} + \frac{10L^2\gamma^2}{\rho^2} \leq c \leq \frac{m^{2/3}}{k^2}$, $m \geq \max\left(\frac{3}{2}, k^3, \frac{8^{3/2}}{(3k)^{3/2}}\right)$ and $k > 0$, we have*

$$
\frac{1}{T} \sum_{t=1}^{T} \mathbb{E}\|\mathcal{G}_{\mathcal{X}}(x_t, \nabla f(x_t), \gamma)\| \leq \frac{1}{T} \sum_{t=1}^{T} \mathbb{E}[\mathcal{M}_t] \leq \frac{2\sqrt{2G}m^{1/6}}{T^{1/2}} + \frac{2\sqrt{2G}}{T^{1/3}},
\tag{37}
$$

*where $G = \frac{f(x_1) - f^*}{k\rho\gamma} + \frac{m^{1/3}\sigma^2}{8k^2L^2\gamma^2} + \frac{k^2c^2\sigma^2}{4L^2\gamma^2} \ln(m + T)$.*

*Proof.* Since $\mu_t = \frac{k}{(m+t)^{1/3}}$ on $t$ is decreasing and $m \geq k^3$, we have $\mu_t \leq \mu_0 = \frac{k}{m^{1/3}} \leq 1$ for all $t \geq 0$. Due to $c \leq \frac{m^{2/3}}{k^2}$, we have $\alpha_{t+1} = c\mu_t^2 \leq c\mu_0^2 \leq \frac{ck^2}{m^{2/3}} \leq \frac{m^{2/3}}{k^2} \frac{k^2}{m^{2/3}} = 1$. Considering $0 < \gamma \leq \frac{\rho m^{1/3}}{4Lk}$, we have $0 < \gamma \leq \frac{\rho m^{1/3}}{4Lk} \leq \frac{\rho m^{1/3}}{2Lk} = \frac{\rho}{2L\mu_0} \leq \frac{\rho}{2L\mu_t}$ for any $t \geq 0$. Thus, the parameters $\mu_t$, $\alpha_{t+1}$ for all $t \geq 0$ and $\gamma$ satisfy the conditions in the above Lemmas 3 and 4.

According to the concavity of the function $f(x) = x^{1/3}$, *i.e.*, $(x+y)^{1/3} \leq x^{1/3} + \frac{y}{3x^{2/3}}$ for any $x, y > 0$, we can obtain

$$
\begin{aligned}
\frac{1}{\mu_t} - \frac{1}{\mu_{t-1}} &= \frac{1}{k}\big((m+t)^{\frac{1}{3}} - (m+t-1)^{\frac{1}{3}}\big) \\
&\leq \frac{1}{3k(m+t-1)^{2/3}} \leq \frac{1}{3k(m/3+t)^{2/3}} \\
&\leq \frac{3^{2/3}}{3k(m+t)^{2/3}} = \frac{3^{2/3}}{3k^3}\frac{k^2}{(m+t)^{2/3}} \\
&= \frac{3^{2/3}}{3k^3}\mu_t^2 \leq \frac{1}{k^3}\mu_t,
\end{aligned}
\tag{38}
$$

where the second inequality holds by $m \geq 3/2$ and the last inequality is due to $0 < \mu_t \leq 1$.

According to Lemma 4, we have

$$
\begin{aligned}
&\frac{1}{\mu_t}\mathbb{E}\|\nabla f(x_{t+1}) - g_{t+1}\|^2 - \frac{1}{\mu_{t-1}}\mathbb{E}\|\nabla f(x_t) - g_t\|^2 \\
&\leq \big(\frac{(1-\alpha_{t+1})^2}{\mu_t} - \frac{1}{\mu_{t-1}}\big)\mathbb{E}\|\nabla f(x_t) - g_t\|^2 + 2(1-\alpha_{t+1})^2 L^2\mu_t\mathbb{E}\|\tilde{x}_{t+1} - x_t\|^2 + \frac{2\alpha_{t+1}^2\sigma^2}{\mu_t} \\
&\leq \big(\frac{1-\alpha_{t+1}}{\mu_t} - \frac{1}{\mu_{t-1}}\big)\mathbb{E}\|\nabla f(x_t) - g_t\|^2 + 2L^2\mu_t\mathbb{E}\|\tilde{x}_{t+1} - x_t\|^2 + \frac{2\alpha_{t+1}^2\sigma^2}{\mu_t} \\
&= \big(\frac{1}{\mu_t} - \frac{1}{\mu_{t-1}} - c\mu_t\big)\mathbb{E}\|\nabla f(x_t) - g_t\|^2 + 2L^2\mu_t\mathbb{E}\|\tilde{x}_{t+1} - x_t\|^2 + 2c^2\mu_t^3\sigma^2,
\end{aligned}
\tag{39}
$$

where the second inequality is due to $0 < \alpha_{t+1} \leq 1$, and the last equality holds by $\alpha_{t+1} = c\mu_t^2$. Since $c \geq \frac{1}{k^3} + \frac{10L^2\gamma^2}{\rho^2}$, we have

$$
\begin{aligned}
&\frac{1}{\mu_t}\mathbb{E}\|\nabla f(x_{t+1}) - g_{t+1}\|^2 - \frac{1}{\mu_{t-1}}\mathbb{E}\|\nabla f(x_t) - g_t\|^2 \\
&\leq -\frac{10L^2\gamma^2}{\rho^2}\mu_t\mathbb{E}\|\nabla f(x_t) - g_t\|^2 + 2L^2\mu_t\mathbb{E}\|\tilde{x}_{t+1} - x_t\|^2 + 2c^2\mu_t^3\sigma^2.
\end{aligned}
\tag{40}
$$

Due to $c \leq \frac{m^{2/3}}{k^2}$, $c \geq \frac{1}{k^3} + \frac{10L^2\gamma^2}{\rho^2}$ and $0 < \gamma \leq \frac{\rho m^{1/3}}{4kL}$, we require

$$
\frac{1}{k^3} + \frac{10L^2\gamma^2}{\rho^2} \leq \frac{1}{k^3} + \frac{10L^2}{\rho^2}\frac{\rho^2 m^{2/3}}{16k^2L^2} = \frac{1}{k^3} + \frac{5m^{2/3}}{8k^2} \leq \frac{m^{2/3}}{k^2}.
\tag{41}
$$

Then we obtain $m \geq \frac{8^{3/2}}{(3k)^{3/2}}$. In the other words, we need $m \geq \frac{8^{3/2}}{(3k)^{3/2}}$ to ensure $\frac{1}{k^3} + \frac{10L^2\gamma^2}{\rho^2} \leq c \leq \frac{m^{2/3}}{k^2}$.

Next, we define a useful Lyapunov function $\Phi_t = \mathbb{E}\big[f(x_t) + \frac{\rho}{8L^2\gamma}\frac{1}{\mu_{t-1}}\|\nabla f(x_t) - g_t\|^2\big]$ for any $t \geq 1$. Then we have

$$
\begin{aligned}
&\Phi_{t+1} - \Phi_t \\
&= \mathbb{E}\big[f(x_{t+1}) - f(x_t)\big] + \frac{\rho}{8L^2\gamma}\big(\frac{1}{\mu_t}\mathbb{E}\|\nabla f(x_{t+1}) - g_{t+1}\|^2 - \frac{1}{\mu_{t-1}}\mathbb{E}\|\nabla f(x_t) - g_t\|^2\big) \\
&\leq \frac{\gamma\mu_t}{\rho}\mathbb{E}\|\nabla f(x_t) - g_t\|^2 - \frac{\rho\mu_t}{2\gamma}\mathbb{E}\|\tilde{x}_{t+1} - x_t\|^2 - \frac{5\gamma\mu_t}{4\rho}\mathbb{E}\|\nabla f(x_t) - g_t\|^2 + \frac{\rho\mu_t}{4\gamma}\mathbb{E}\|\tilde{x}_{t+1} - x_t\|^2 + \frac{\rho c^2\mu_t^3\sigma^2}{4L^2\gamma} \\
&\leq -\frac{\gamma\mu_t}{4\rho}\mathbb{E}\|\nabla f(x_t) - g_t\|^2 - \frac{\rho\mu_t}{4\gamma}\mathbb{E}\|\tilde{x}_{t+1} - x_t\|^2 + \frac{\rho c^2\eta_t^3\sigma^2}{4L^2\gamma},
\end{aligned}
\tag{42}
$$

where the first inequality is due to the above inequality (40) and the above Lemma 3.

By using the above inequality (42), we have

$$
\frac{1}{T}\sum_{t=1}^{T}\mathbb{E}\big[\frac{\gamma\mu_t}{4\rho}\|\nabla f(x_t)-g_t\|^2+\frac{\rho\mu_t}{4\gamma}\|\tilde{x}_{t+1}-x_t\|^2\big]
$$

$$
\leq \frac{\Phi_1-\Phi_{T+1}}{T}+\frac{1}{T}\sum_{t=1}^{T}\frac{\rho c^2\eta_t^3\sigma^2}{4L^2\gamma}
$$

$$
= \frac{f(x_1)-\mathbb{E}\big[f(x_{T+1})\big]}{T}+\frac{\rho\|\nabla f(x_1)-g_1\|^2}{8L^2\gamma\mu_0 T}-\frac{\rho\mathbb{E}\|\nabla f(x_{T+1})-g_{T+1}\|^2}{8L^2\gamma\mu_T T}+\frac{1}{T}\sum_{t=1}^{T}\frac{\rho c^2\eta_t^3\sigma^2}{4L^2\gamma}
$$

$$
\leq \frac{f(x_1)-f^*}{T}+\frac{\rho\sigma^2}{8L^2\mu_0\gamma T}+\frac{1}{T}\sum_{t=1}^{T}\frac{\rho c^2\sigma^2}{4L^2\gamma}\eta_t^3, \tag{43}
$$

where the last inequality holds by Assumptions 1 and 2. Since $\mu_t=\frac{k}{(m+t)^{1/3}}$ on $t$ is not increasing, we have

$$
\frac{1}{T}\sum_{t=1}^{T}\mathbb{E}\big[\frac{1}{4\rho^2}\|\nabla f(x_t)-g_t\|^2+\frac{1}{4\gamma^2}\|\tilde{x}_{t+1}-x_t\|^2\big]
$$

$$
\leq \frac{f(x_1)-f^*}{T\rho\gamma\mu_T}+\frac{\sigma^2}{8\gamma^2\mu_T\mu_0 L^2 T}+\frac{c^2\sigma^2}{4L^2\gamma^2\mu_T T}\sum_{t=1}^{T}\mu_t^3
$$

$$
\leq \frac{f(x_1)-f^*}{T\rho\gamma\mu_T}+\frac{\sigma^2}{8\gamma^2\mu_T\mu_0 L^2 T}+\frac{c^2\sigma^2}{4L^2\gamma^2\mu_T T}\int_{1}^{T}\frac{k^3}{m+t}dt
$$

$$
\leq \frac{f(x_1)-f^*}{T\rho\gamma\mu_T}+\frac{\sigma^2}{8\gamma^2\mu_T\mu_0 L^2 T}+\frac{c^2k^3\sigma^2}{4L^2\gamma^2\mu_T T}\ln(m+T)
$$

$$
= \Big(\frac{f(x_1)-f^*}{\rho\gamma k}+\frac{m^{1/3}\sigma^2}{8L^2k^2\gamma^2}+\frac{k^2c^2\sigma^2}{4L^2\gamma^2}\ln(m+T)\Big)\frac{(m+T)^{1/3}}{T}, \tag{44}
$$

where the second inequality is due to $\sum_{t=1}^{T}\mu_t^3 dt\leq\int_{1}^{T}\mu_t^3 dt$. Let $G=\frac{f(x_1)-f^*}{k\rho\gamma}+\frac{m^{1/3}\sigma^2}{8k^2L^2\gamma^2}+\frac{k^2c^2\sigma^2}{4L^2\gamma^2}\ln(m+T)$, we have

$$
\frac{1}{T}\sum_{t=1}^{T}\mathbb{E}\big[\frac{1}{4\rho^2}\|\nabla f(x_t)-g_t\|^2+\frac{1}{4\gamma^2}\|\tilde{x}_{t+1}-x_t\|^2\big]\leq\frac{G(m+T)^{1/3}}{T}. \tag{45}
$$

According to Jensen's inequality, we have

$$
\frac{1}{T}\sum_{t=1}^{T}\mathbb{E}\big[\frac{1}{2\rho}\|\nabla f(x_t)-g_t\|+\frac{1}{2\gamma}\|\tilde{x}_{t+1}-x_t\|\big]
$$

$$
\leq \big(\frac{2}{T}\sum_{t=1}^{T}\mathbb{E}\big[\frac{1}{4\rho^2}\|\nabla f(x_t)-g_t\|^2+\frac{1}{4\gamma^2}\|\tilde{x}_{t+1}-x_t\|^2\big]\big)^{1/2}
$$

$$
\leq \frac{\sqrt{2G}}{T^{1/2}}(m+T)^{1/6}\leq\frac{\sqrt{2G}m^{1/6}}{T^{1/2}}+\frac{\sqrt{2G}}{T^{1/3}}, \tag{46}
$$

where the last inequality holds by $(a+b)^{1/6}\leq a^{1/6}+b^{1/6}$ for any $a,b>0$. Thus we can obtain

$$
\frac{1}{T}\sum_{t=1}^{T}\mathbb{E}\big[\mathcal{M}_t\big]=\frac{1}{T}\sum_{t=1}^{T}\mathbb{E}\big[\frac{1}{\rho}\|\nabla f(x_t)-g_t\|+\frac{1}{\gamma}\|\tilde{x}_{t+1}-x_t\|\big]\leq\frac{2\sqrt{2G}m^{1/6}}{T^{1/2}}+\frac{2\sqrt{2G}}{T^{1/3}}. \tag{47}
$$

Let $w_t(x)=\frac{1}{2}x^T H_t x$, we give a prox-function (i.e., Bregman distance) [4, 5, 15] associated with $w_t(x)$, defined as:

$$
V_t(x,x_t)=w_t(x)-\big[w_t(x_t)+\langle\nabla w_t(x_t),x-x_t\rangle\big]=\frac{1}{2}(x-x_t)^T H_t(x-x_t). \tag{48}
$$

Then the step 9 of Algorithm 1 is equivalent to the following generalized projection problem:

$$
\tilde{x}_{t+1}=\arg\min_{x\in\mathcal{X}}\big\{\langle g_t,x\rangle+\frac{1}{\gamma}V_t(x,x_t)\big\}. \tag{49}
$$

Let $\mathcal{G}_{\mathcal{X}}(x_t,g_t,\gamma)=\frac{1}{\gamma}(x_t-\tilde{x}_{t+1})$. As in [15], we define a gradient mapping $\mathcal{G}_{\mathcal{X}}(x_t,\nabla f(x_t),\gamma)=\frac{1}{\gamma}(x_t-x_{t+1}^+)$, where

$$
x_{t+1}^+=\arg\min_{x\in\mathcal{X}}\big\{\langle\nabla f(x_t),x\rangle+\frac{1}{\gamma}V_t(x,x_t)\big\}. \tag{50}
$$

According to the above Lemma 2, we have $\|\mathcal{G}_{\mathcal{X}}(x_t, g_t, \gamma) - \mathcal{G}_{\mathcal{X}}(x_t, \nabla f(x_t), \gamma)\| \leq \frac{1}{\rho}\|\nabla f(x_t) - g_t\|$. Then we have

$$
\begin{aligned}
\|\mathcal{G}_{\mathcal{X}}(x_t, \nabla f(x_t), \gamma)\| &\leq \|\mathcal{G}_{\mathcal{X}}(x_t, g_t, \gamma)\| + \|\mathcal{G}_{\mathcal{X}}(x_t, g_t, \gamma) - \mathcal{G}_{\mathcal{X}}(x_t, \nabla f(x_t), \gamma)\| \\
&\leq \|\mathcal{G}_{\mathcal{X}}(x_t, g_t, \gamma)\| + \frac{1}{\rho}\|\nabla f(x_t) - g_t\| \\
&= \frac{1}{\gamma}\|x_t - \tilde{x}_{t+1}\| + \frac{1}{\rho}\|\nabla f(x_t) - g_t\| = \mathcal{M}_t.
\end{aligned}
\tag{51}
$$

By combining the above inequalities (47) with (51), we have

$$
\frac{1}{T}\sum_{t=1}^{T}\mathbb{E}\|\mathcal{G}_{\mathcal{X}}(x_t, \nabla f(x_t), \gamma)\| \leq \frac{1}{T}\sum_{t=1}^{T}\mathbb{E}[\mathcal{M}_t] \leq \frac{2\sqrt{2G}m^{1/6}}{T^{1/2}} + \frac{2\sqrt{2G}}{T^{1/3}}.
\tag{52}
$$

$\square$

**Corollary 3.** *(Restatement of Corollary 1) In Algorithm 1, under the above Assumptions (1,2,3,4), when $\mathcal{X} = \mathbb{R}^d$, and given $\tau = 1$, $\mu_t = \frac{k}{(m+t)^{1/3}}$ and $\alpha_{t+1} = c\mu_t^2$ for all $t \geq 0$, $\gamma = \frac{\rho m^{1/3}}{\nu k L}$ ($\nu \geq 4$), $\frac{1}{k^3} + \frac{10L^2\gamma^2}{\rho^2} \leq c \leq \frac{m^{2/3}}{k^2}$, $m \geq \max\left(\frac{3}{2}, k^3, \frac{8^{3/2}}{(3k)^{3/2}}\right)$ and $k > 0$, we have*

$$
\frac{1}{T}\sum_{t=1}^{T}\mathbb{E}\|\nabla f(x_t)\| \leq \frac{\max_{1 \leq t \leq T}\|H_t\|}{\rho}\left(\frac{2\sqrt{2G'}}{T^{1/2}} + \frac{2\sqrt{2G'}}{m^{1/6}T^{1/3}}\right),
\tag{53}
$$

*where $G' = \nu L(f(x_1) - f^*) + \frac{\nu^2\sigma^2}{8} + \frac{\nu^2 k^4 c^2\sigma^2}{4m^{1/3}}\ln(m+T)$.*

*Proof.* When $\mathcal{X} = \mathbb{R}^d$, the step 9 of Algorithm 1 is equivalent to $\tilde{x}_{t+1} = x_t - \gamma H_t^{-1}g_t$. Then we have

$$
\begin{aligned}
\mathcal{M}_t &= \frac{1}{\rho}\|\nabla f(x_t) - g_t\| + \|H_t^{-1}g_t\| = \frac{1}{\rho}\|\nabla f(x_t) - g_t\| + \frac{1}{\|H_t\|}\|H_t\|\|H_t^{-1}g_t\| \\
&\geq \frac{1}{\rho}\|\nabla f(x_t) - g_t\| + \frac{1}{\|H_t\|}\|g_t\| \\
&\geq \frac{1}{\|H_t\|}\left(\|\nabla f(x_t) - g_t\| + \|g_t\|\right) \geq \frac{1}{\|H_t\|}\|\nabla f(x_t)\|,
\end{aligned}
\tag{54}
$$

where the second last inequality holds by $\|H_t\| \geq \rho$. Under the same conditions in Theorem 3, according to the above inequality (47), we can obtain

$$
\frac{1}{T}\sum_{t=1}^{T}\mathbb{E}\left[\frac{1}{\|H_t\|}\|\nabla f(x_t)\|\right] \leq \frac{1}{T}\sum_{t=1}^{T}\mathbb{E}[\mathcal{M}_t] \leq \frac{2\sqrt{2G}m^{1/6}}{T^{1/2}} + \frac{2\sqrt{2G}}{T^{1/3}}.
\tag{55}
$$

Since $\gamma = \frac{\rho m^{1/3}}{\nu k L}$ ($\nu \geq 4$), we have

$$
\begin{aligned}
G &= \frac{f(x_1) - f^*}{k\rho\gamma} + \frac{m^{1/3}\sigma^2}{8k^2 L^2\gamma^2} + \frac{k^2 c^2\sigma^2}{4L^2\gamma^2}\ln(m+T) \\
&= \frac{\nu L(f(x_1) - f^*)}{\rho^2 m^{1/3}} + \frac{\nu^2\sigma^2}{8\rho^2 m^{1/3}} + \frac{\nu^2 k^4 c^2\sigma^2}{4\rho^2 m^{2/3}}\ln(m+T) \\
&= \frac{1}{\rho^2 m^{1/3}}G',
\end{aligned}
\tag{56}
$$

where $G' = \nu L(f(x_1) - f^*) + \frac{\nu^2\sigma^2}{8} + \frac{\nu^2 k^4 c^2\sigma^2}{4m^{1/3}}\ln(m+T)$.

Plugging $G = \frac{1}{\rho^2 m^{1/3}}G'$ into the above inequality (55), we obtain

$$
\frac{1}{T}\sum_{t=1}^{T}\mathbb{E}\|\nabla f(x_t)\| \leq \frac{\max_{1 \leq t \leq T}\|H_t\|}{\rho}\left(\frac{2\sqrt{2G'}}{T^{1/2}} + \frac{2\sqrt{2G'}}{m^{1/6}T^{1/3}}\right).
\tag{57}
$$

$\square$

## A.2 Convergence Analysis of SUPER-ADAM ($\tau = 0$)

In this subsection, we provide the convergence analysis of our SUPER-ADAM ($\tau = 0$) algorithm.

**Lemma 5.** *In Algorithm 1, given $\tau = 0$ and $0 < \alpha_{t+1} \leq 1$ for all $t \geq 0$, we have*

$$\mathbb{E}\|\nabla f(x_{t+1}) - g_{t+1}\|^2 \leq (1 - \alpha_{t+1})\mathbb{E}\|\nabla f(x_t) - g_t\|^2 + \frac{1}{\alpha_{t+1}}L^2\mu_t^2\mathbb{E}\|\tilde{x}_{t+1} - x_t\|^2 + \alpha_{t+1}^2\sigma^2. \quad (58)$$

*Proof.* By the definition of $g_{t+1}$ in Algorithm 1 with $\tau = 0$, we have $g_{t+1} = (1 - \alpha_{t+1})g_t + \alpha_{t+1}\nabla f(x_{t+1}; \xi_{t+1})$. Then we have

$$\mathbb{E}\|\nabla f(x_{t+1}) - g_{t+1}\|^2$$
$$= \mathbb{E}\|\nabla f(x_t) - g_t + \nabla f(x_{t+1}) - \nabla f(x_t) - (g_{t+1} - g_t)\|^2$$
$$= \mathbb{E}\|\nabla f(x_t) - g_t + \nabla f(x_{t+1}) - \nabla f(x_t) + \alpha_{t+1}g_t - \alpha_{t+1}\nabla f(x_{t+1}; \xi_{t+1})\|^2$$
$$= \mathbb{E}\|\alpha_{t+1}(\nabla f(x_{t+1}) - \nabla f(x_{t+1}; \xi_{t+1})) + (1 - \alpha_{t+1})(\nabla f(x_t) - g_t) + (1 - \alpha_{t+1})(\nabla f(x_{t+1}) - \nabla f(x_t))\|^2$$
$$= \alpha_{t+1}^2\mathbb{E}\|\nabla f(x_{t+1}) - \nabla f(x_{t+1}; \xi_{t+1})\|^2 + (1 - \alpha_{t+1})^2\mathbb{E}\|\nabla f(x_t) - g_t + \nabla f(x_{t+1}) - \nabla f(x_t)\|^2$$
$$\leq \alpha_{t+1}^2\mathbb{E}\|\nabla f(x_{t+1}) - \nabla f(x_{t+1}; \xi_{t+1})\|^2 + (1 - \alpha_{t+1})^2(1 + \alpha_{t+1})\mathbb{E}\|\nabla f(x_t) - g_t\|^2$$
$$\quad + (1 - \alpha_{t+1})^2(1 + \frac{1}{\alpha_{t+1}})\mathbb{E}\|\nabla f(x_{t+1}) - \nabla f(x_t)\|^2$$
$$\leq (1 - \alpha_{t+1})\mathbb{E}\|\nabla f(x_t) - g_t\|^2 + \frac{1}{\alpha_{t+1}}\mathbb{E}\|\nabla f(x_{t+1}) - \nabla f(x_t)\|^2 + \alpha_{t+1}^2\mathbb{E}\|\nabla f(x_{t+1}) - \nabla f(x_{t+1}; \xi_{t+1})\|^2$$
$$\leq (1 - \alpha_{t+1})\mathbb{E}\|\nabla f(x_t) - g_t\|^2 + \frac{1}{\alpha_{t+1}}L^2\mu_t^2\mathbb{E}\|\tilde{x}_{t+1} - x_t\|^2 + \alpha_{t+1}^2\sigma^2, \quad (59)$$

where the fourth equality holds by $\mathbb{E}_{\xi_{t+1}}[\nabla f(x_{t+1}; \xi_{t+1}) - \nabla f(x_{t+1})] = 0$ and $\xi_{t+1}$ is independent on variables $x_t$ and $x_{t+1}$; the first inequality holds by Young's inequality; the second inequality is due to $0 < \alpha_{t+1} \leq 1$ such that $(1 - \alpha_{t+1})^2(1 + \alpha_{t+1}) = 1 - \alpha_{t+1} - \alpha_{t+1}^2 + \alpha_{t+1}^3 \leq 1 - \alpha_{t+1}$ and $(1 - \alpha_{t+1})^2(1 + \frac{1}{\alpha_{t+1}}) \leq (1 - \alpha_{t+1})(1 + \frac{1}{\alpha_{t+1}}) = -\alpha_{t+1} + \frac{1}{\alpha_{t+1}} \leq \frac{1}{\alpha_{t+1}}$; the last inequality is due to Assumptions 1, 5, and and $x_{t+1} = x_t + \mu_t(\tilde{x}_{t+1} - x_t)$. $\qquad \square$

**Theorem 4.** *(Restatement of Theorem 2) In Algorithm 1, under the above Assumptions (1,2,3,5), when $\mathcal{X} \subset \mathbb{R}^d$, and given $\tau = 0$, $\mu_t = \frac{k}{(m+t)^{1/2}}$, $\alpha_{t+1} = c\mu_t$ for all $t \geq 0$, $k > 0$, $0 < \gamma \leq \frac{\rho m^{1/2}}{8Lk}$, $\frac{8L\gamma}{\rho} \leq c \leq \frac{m^{1/2}}{k}$, and $m \geq k^2$, we have*

$$\frac{1}{T}\sum_{t=1}^{T}\mathbb{E}\|\mathcal{G}_{\mathcal{X}}(x_t, \nabla f(x_t), \gamma)\| \leq \frac{1}{T}\sum_{t=1}^{T}\mathbb{E}[\mathcal{M}_t] \leq \frac{2\sqrt{2M}m^{1/4}}{T^{1/2}} + \frac{2\sqrt{2M}}{T^{1/4}},$$

*where $M = \frac{f(x_1) - f^*}{\rho\gamma k} + \frac{2\sigma^2}{\rho\gamma kL} + \frac{2m\sigma^2}{\rho\gamma kL}\ln(m + T)$.*

*Proof.* Since $\mu_t = \frac{k}{(m+t)^{1/2}}$ is decreasing on $t$ and $m \geq k^2$, we have $\mu_t \leq \mu_0 = \frac{k}{m^{1/2}} \leq 1$ for all $t \geq 0$. Due to $c \leq \frac{m^{1/2}}{k}$, we have $\alpha_{t+1} = c\mu_t \leq c\mu_0 \leq \frac{m^{1/2}}{k}\frac{k}{m^{1/2}} = 1$ for all $t \geq 0$. Since $0 < \gamma \leq \frac{\rho m^{1/2}}{8Lk}$, we have $\gamma \leq \frac{\rho m^{1/2}}{8Lk} \leq \frac{\rho m^{1/2}}{2Lk} = \frac{\rho}{2L\mu_0} \leq \frac{\rho}{2L\mu_t}$ for all $t \geq 0$. Thus, the parameters $\mu_t$, $\alpha_{t+1}$ for all $t \geq 0$ and $\gamma$ satisfy the conditions in the above Lemmas 3 and 5.

According to Lemma 5, we have

$$\mathbb{E}\|\nabla f(x_{t+1}) - g_{t+1}\|^2 - \mathbb{E}\|\nabla f(x_t) - g_t\|^2$$
$$\leq -\alpha_{t+1}\mathbb{E}\|\nabla f(x_t) - g_t\|^2 + \frac{1}{\alpha_{t+1}}L^2\mu_t^2\mathbb{E}\|\tilde{x}_{t+1} - x_t\|^2 + \alpha_{t+1}^2\sigma^2$$
$$= -c\mu_t\mathbb{E}\|\nabla f(x_t) - g_t\|^2 + \frac{L^2\mu_t}{c}\mathbb{E}\|\tilde{x}_{t+1} - x_t\|^2 + c^2\mu_t^2\sigma^2$$
$$\leq -\frac{8L\gamma\mu_t}{\rho}\mathbb{E}\|\nabla f(x_t) - g_t\|^2 + \frac{L\rho\mu_t}{8\gamma}\mathbb{E}\|\tilde{x}_{t+1} - x_t\|^2 + \frac{m\mu_t^2\sigma^2}{k^2}, \quad (60)$$

where the first equality is due to $\alpha_{t+1} = c\mu_t$ and the last equality holds by $\frac{8L\gamma}{\rho} \leq c \leq \frac{m^{1/2}}{k}$.

Next, we define a *Lyapunov* function $\Omega_t = \mathbb{E}\big[f(x_t) + \frac{2}{L}\|\nabla f(x_t) - g_t\|^2\big]$ for any $t \geq 1$. Then we have

$$\Omega_{t+1} - \Omega_t = \mathbb{E}\big[f(x_{t+1}) - f(x_t) + \frac{2}{L}\big(\|\nabla f(x_{t+1}) - g_{t+1}\|^2 - \|\nabla f(x_t) - g_t\|^2\big)\big]$$

$$\leq \frac{\gamma\mu_t}{\rho}\mathbb{E}\|\nabla f(x_t) - g_t\|^2 - \frac{\rho\mu_t}{2\gamma}\mathbb{E}\|\tilde{x}_{t+1} - x_t\|^2 - \frac{16\gamma\mu_t}{\rho}\mathbb{E}\|\nabla f(x_t) - g_t\|^2$$

$$+ \frac{\rho\mu_t}{4\gamma}\mathbb{E}\|\tilde{x}_{k+1} - x_t\|^2 + \frac{2m\mu_t^2\sigma^2}{k^2 L}$$

$$\leq -\frac{\gamma\mu_t}{4\rho}\mathbb{E}\|\nabla f(x_t) - g_t\|^2 - \frac{\rho\mu_t}{4\gamma}\mathbb{E}\|\tilde{x}_{t+1} - x_t\|^2 + \frac{2m\mu_t^2\sigma^2}{k^2 L}, \tag{61}$$

where the first inequality follows by the above inequality (60) and the above Lemma 3.

Taking average over $t = 1, 2, \cdots, T$ on both sides of (61), we have

$$\frac{1}{T}\sum_{t=1}^{T}\mathbb{E}[\frac{\gamma\mu_t}{4\rho}\|\nabla f(x_t) - g_t\|^2 + \frac{\rho\mu_t}{4\gamma}\|\tilde{x}_{t+1} - x_t\|^2]$$

$$\leq \frac{\Omega_1 - \Omega_{T+1}}{T} + \frac{1}{T}\sum_{t=1}^{T}\frac{2m\mu_t^2\sigma^2}{k^2 L}$$

$$= \frac{f(x_1) - \mathbb{E}[f(x_{T+1})]}{T} + \frac{2}{LT}\|\nabla f(x_1) - g_1\|^2 - \frac{2}{LT}\mathbb{E}\|\nabla f(x_{T+1}) - g_{T+1}\|^2 + \frac{2m\sigma^2}{Tk^2 L}\sum_{t=1}^{T}\mu_t^2$$

$$\leq \frac{f(x_1) - f^*}{T} + \frac{2\sigma^2}{LT} + \frac{2m\sigma^2}{Tk^2 L}\sum_{t=1}^{T}\mu_t^2$$

$$\leq \frac{f(x_1) - f^*}{T} + \frac{2\sigma^2}{LT} + \frac{2m\sigma^2}{Tk^2 L}\int_{t=1}^{T}\frac{k^2}{m+t}dt$$

$$\leq \frac{f(x_1) - f^*}{T} + \frac{2\sigma^2}{LT} + \frac{2m\sigma^2}{TL}\ln(m+T), \tag{62}$$

where the second inequality holds by Assumptions 1 and 2. Since $\mu_t = \frac{k}{(m+t)^{1/2}}$ is decreasing on $t$, we have

$$\frac{1}{T}\sum_{t=1}^{T}\mathbb{E}[\frac{1}{4\rho^2}\|\nabla f(x_t) - g_t\|^2 + \frac{1}{4\gamma^2}\|\tilde{x}_{t+1} - x_t\|^2]$$

$$\leq \frac{f(x_1) - f^*}{\rho\gamma\mu_T T} + \frac{2\sigma^2}{\rho\gamma\mu_T LT} + \frac{2m\sigma^2}{\rho\gamma\mu_T TL}\ln(m+T)$$

$$= \Big(\frac{f(x_1) - f^*}{\rho\gamma k} + \frac{2\sigma^2}{\rho\gamma kL} + \frac{2m\sigma^2}{\rho\gamma kL}\ln(m+T)\Big)\frac{(m+T)^{\frac{1}{2}}}{T}. \tag{63}$$

Let $M = \frac{f(x_1) - f^*}{\rho\gamma k} + \frac{2\sigma^2}{\rho\gamma kL} + \frac{2m\sigma^2}{\rho\gamma kL}\ln(m+T)$, the above inequality (63) reduces to

$$\frac{1}{T}\sum_{t=1}^{T}\mathbb{E}[\frac{1}{4\rho^2}\|\nabla f(x_t) - g_t\|^2 + \frac{1}{4\gamma^2}\|\tilde{x}_{t+1} - x_t\|^2] \leq \frac{M}{T}(m+T)^{\frac{1}{2}}. \tag{64}$$

According to Jensen's inequality, we have

$$\frac{1}{T}\sum_{t=1}^{T}\mathbb{E}[\frac{1}{2\rho}\|\nabla f(x_t) - g_t\| + \frac{1}{2\gamma}\|\tilde{x}_{t+1} - x_t\|]$$

$$\leq \Big(\frac{2}{T}\sum_{t=1}^{T}\mathbb{E}\big[\frac{1}{4\rho^2}\|\nabla f(x_t) - g_t\|^2 + \frac{1}{4\gamma^2}\|\tilde{x}_{t+1} - x_t\|^2\big]\Big)^{1/2}$$

$$\leq \frac{\sqrt{2M}}{T^{1/2}}(m+T)^{1/4} \leq \frac{\sqrt{2M}m^{1/4}}{T^{1/2}} + \frac{\sqrt{2M}}{T^{1/4}}, \tag{65}$$

where the last inequality is due to the inequality $(a+b)^{1/4} \leq a^{1/4} + b^{1/4}$ for all $a, b \geq 0$. Thus, we have

$$\frac{1}{T}\sum_{t=1}^{T}\mathbb{E}[\mathcal{M}_t] = \frac{1}{T}\sum_{t=1}^{T}\mathbb{E}[\frac{1}{\rho}\|\nabla f(x_t) - g_t\| + \frac{1}{\gamma}\|\tilde{x}_{t+1} - x_t\|] \leq \frac{2\sqrt{2M}m^{1/4}}{T^{1/2}} + \frac{2\sqrt{2M}}{T^{1/4}}. \tag{66}$$

By using the above inequality (51), we obtain

$$\frac{1}{T}\sum_{t=1}^{T}\mathbb{E}\|\mathcal{G}_{\mathcal{X}}(x_t, \nabla f(x_t), \gamma)\| \leq \frac{1}{T}\sum_{t=1}^{T}\mathbb{E}[\mathcal{M}_t] \leq \frac{2\sqrt{2M}m^{1/4}}{T^{1/2}} + \frac{2\sqrt{2M}}{T^{1/4}}. \tag{67}$$

$\square$

**Corollary 4.** *(Restatement of Corollary 2) In Algorithm 1, under the above Assumptions (**1,2,3,5**), when* $\mathcal{X} = \mathbb{R}^d$, *and given* $\tau = 0$, $\mu_t = \frac{k}{(m+t)^{1/2}}$, $\alpha_{t+1} = c\mu_t$ *for all* $t \geq 0$, $k > 0$, $\gamma = \frac{\rho m^{1/2}}{\nu L k}$ $(\nu \geq 8)$, $\frac{8L\gamma}{\rho} \leq c \leq \frac{m^{1/2}}{k}$, *and* $m \geq k^2$, *we have*

$$\frac{1}{T}\sum_{t=1}^{T}\mathbb{E}\|\nabla f(x_t)\| \leq \frac{\max_{1 \leq t \leq T}\|H_t\|}{\rho}\left(\frac{2\sqrt{2M'}}{T^{1/2}} + \frac{2\sqrt{2M'}}{m^{1/4}T^{1/4}}\right),$$

*where* $M' = \nu L(f(x_1) - f^*) + 2\nu\sigma^2 + 2\nu m\sigma^2 \ln(m+T)$.

*Proof.* Under the same conditions of Theorem 4, according to the above inequalities (54) and (66), we can obtain

$$\frac{1}{T}\sum_{t=1}^{T}\mathbb{E}\Big[\frac{1}{\|H_t\|}\|\nabla f(x_t)\|\Big] \leq \frac{1}{T}\sum_{t=1}^{T}\mathbb{E}[\mathcal{M}_t] \leq \frac{2\sqrt{2M}m^{1/4}}{T^{1/2}} + \frac{2\sqrt{2M}}{T^{1/4}}. \tag{68}$$

Since $\gamma = \frac{\rho m^{1/2}}{\nu L k}$ $(\nu \geq 8)$, we have

$$\begin{aligned} M &= \frac{f(x_1) - f^*}{\rho\gamma k} + \frac{2\sigma^2}{\rho\gamma kL} + \frac{2m\sigma^2}{\rho\gamma kL}\ln(m+T) \\ &= \frac{\nu L(f(x_1) - f^*)}{\rho^2 m^{1/2}} + \frac{2\nu\sigma^2}{\rho^2 m^{1/2}} + \frac{2\nu m^{1/2}\sigma^2}{\rho^2}\ln(m+T) \\ &= \frac{1}{\rho^2 m^{1/2}}M', \end{aligned} \tag{69}$$

where $M' = \nu L(f(x_1) - f^*) + 2\nu\sigma^2 + 2\nu m\sigma^2 \ln(m+T)$.

Plugging $M = \frac{1}{\rho^2 m^{1/2}}M'$ into the above inequality (68), we obtain

$$\frac{1}{T}\sum_{t=1}^{T}\mathbb{E}\|\nabla f(x_t)\| \leq \frac{\max_{1 \leq t \leq T}\|H_t\|}{\rho}\left(\frac{2\sqrt{2M'}}{T^{1/2}} + \frac{2\sqrt{2M'}}{m^{1/4}T^{1/4}}\right). \tag{70}$$

$\square$

# B   Additional Experimental Results

In the section, we conduct some numerical experiments to empirically evaluate our SUPER-ADAM algorithm on two deep learning tasks as in [25]: **image classification** on CIFAR-10, CIFAR-100 and Image-Net datasets and **language modeling** on Wiki-Text2 dataset (Please see Table 2).

Table 2: Summary of setups in the experiments.

| Task | Architecture | Dataset |
|---|---|---|
| Image Classification | ResNet18 | CIFAR-10 |
| Image Classification | VGG19 | CIFAR-100 |
| Image Classification | ResNet34 | Image-Net |
| Language Modeling | Two-layer LSTM | Wiki-Text2 |
| Language Modeling | Transformer | Wiki-Text2 |

## B.1   Image Classification Task

We add some experimental results of ImageNet data given in Figure 6.

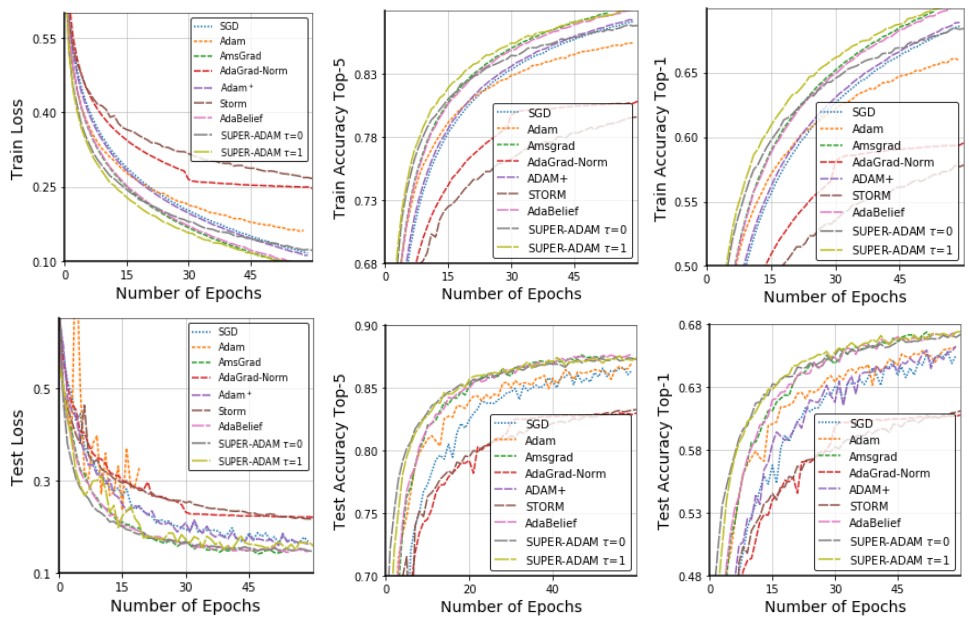

Figure 6: Experimental Results of Image-Net by Different Optimizers over ResNet-34.

## B.2    Language Modeling Task

In the experiment, we conduct language modeling task on the Wiki-Text2 dataset. Specifically, we train a 2-layer LSTM [21] and a 2-layer Transformer over the WiKi-Text2 dataset. For the 2-layer Transformer, we use 200 dimensional word embeddings, 200 hidden unites and 2 heads. What's more, we set the batch size as 20 and trains for 40 epochs with dropout rate 0.5. We also clip the gradients by norm 0.25 same as when we use LSTM. We decrease the learning by 4 whenever the validation error increases. For the learning rate, we also do grid search and report the best one for each optimizer. In Adam and Amsgrad algorithms, we set the learning rate as 0.0002 in Transformer. In AdaGrad-Norm algorithm, the best learning rate is 10 in Transformer. In Adam$^+$ algorithm, we use the learning rate 20. In AdaBelief algorithm, we set the learing 1. In STORM algorithm, we set $k = 12.5$, $w = 100$ and $c = 10$. In our SUPER-ADAM algorithm, we set $k = 1$, $m = 100$, $c = 20$, $\gamma = 0.002$ when $\tau = 1$, while $k = 1$, $m = 100$, $c = 20$, $\gamma = 0.003$ when $\tau = 0$.

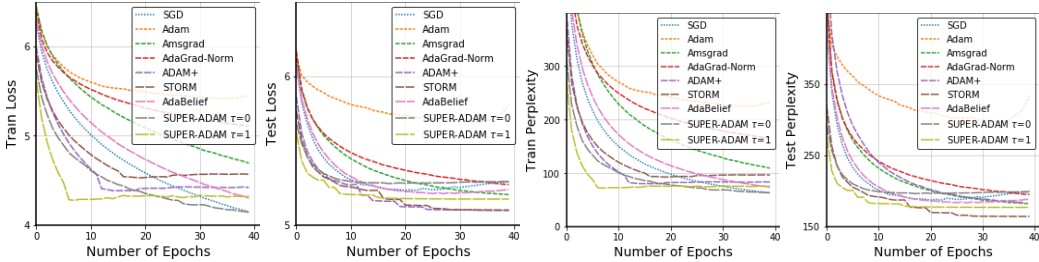

Figure 7: Experimental Results of WikiText-2 by Different Optimizers over Transformer.

Figure 7 shows that both train and test perplexities (losses) for different optimizers. When $\tau = 1$, our SUPER-ADAM algorithm outperforms all the other optimizers. When $\tau = 0$, our SUPER-ADAM optimizer gets a comparable performance with the other Adam-type algorithms.