# OpenReview forum: "SUPER-ADAM: Faster and Universal Framework of Adaptive Gradients"
_NeurIPS.cc/2021/Conference — NeurIPS 2021 Poster_

### Official Review · Reviewer_v7co · 2021-07-15

**Rating:** 8
**Confidence:** 4

**Summary:**

This paper presents a universal adaptive gradient framework (i.e., super-adam), which can flexibly incorporate the momentum and variance reduced techniques to effectively accelerate the convergence of the algorithm. Moreover, it provides a sound convergence analysis framework for the super-adam under the nonconvex setting. Experimental results on some deep learning tasks demonstrate the efficiency of the proposed super-adam algorithm. Super-adam may have broad applications in training deep learning models. Overall, I recommend accepting this paper.



**Limitations And Societal Impact:**

yes

**Main Review:**

I have the following major comments/questions:

1) In the theoretical analysis, the parameter $\rho$ is very important. Could you summarize the impact of this parameter on the convergence analysis? E.g., how it affects the convergence properties of the proposed algorithm?

2) In the super-adam algorithm, we can choose flexible adaptive learning rates. Could you give us a guide on how to choose a good adaptive learning rate in the experiment?

Minor comments:

1) In Abstract, the line 9: “…can flexibly integrates…”  should  be  “… can flexibly integrate…”;

2) In the convergence analysis, the gradient estimator $g_{t+1}$ is right (in the line 436).  At step 13 of the super-adam algorithm,  there exists a symbol error:  $g_t -$ should be  $g_t +$.

3) In the experiment, do you use the same adaptive learning rates in super-adam ($\tau=1$) and super-adam ($\tau=0$) ?

**Time Spent Reviewing:**

3

---

> ### Author Response · Authors · 2021-08-09
> **Responses for comments**
>
>
> Thanks for your positive comments. We address your concerns as follows:
>
> 1) In our convergence results, the parameter $\rho$ mainly affects the tuning parameter selection. For example, the parameters $\gamma$ and $ \alpha$ depend on the parameter $\rho$ (Please see Theorems 1-2). At the same time, $\rho$ also affects our convergence measure.  When $\rho \geq 1$, our convergence measure $\mathcal{M}_t$ is tighter than the gradient mapping measure $|| \mathcal{G}_\mathcal{X} (x_t,\nabla f(x_t),\gamma) ||$ (Please see lines 193-199 at page 6).
>
> Specifically, according to Proposition 1 in [10], we have $||\mathcal{G}_\mathcal{X}(x_t,g_t,\gamma)-\mathcal{G}_\mathcal{X} (x_t,\nabla f(x_t),\gamma)|| \leq \frac{1}{\rho} || \nabla f(x_t)-g_t ||$.
>  Since $||\mathcal{G}_\mathcal{X}(x_t,\nabla f(x_t),\gamma)||  \leq ||\mathcal{G}_\mathcal{X}(x_t,g_t,\gamma)|| + ||\mathcal{G}_\mathcal{X}(x_t,g_t,\gamma)-\mathcal{G}_\mathcal{X}(x_t,\nabla f(x_t),\gamma)||$,
>  we have $||\mathcal{G}_\mathcal{X}(x_t,\nabla f(x_t),\gamma)|| \leq ||\mathcal{G}_\mathcal{X}(x_t,g_t,\gamma)|| + \frac{1}{\rho}||\nabla f(x_t)-g_t|| = \frac{1}{\rho}(  \rho ||\mathcal{G}_\mathcal{X} (x_t,g_t,\gamma)|| + ||\nabla f(x_t) - g_t|| ) = \frac{1}{\rho} \mathcal{M}_t $.
>
> When $\mathcal{M}_{t}   \rightarrow 0$,
> we can obtain $|| \mathcal{G}_\mathcal{X} (x_t,\nabla f(x_t),\gamma) || \rightarrow 0$, where $x_t$ is a stationary point or local minimum of the problem (1).
>
> Clearly, when $\rho \geq 1$, our measure $\mathbb{E}[\mathcal{M}_{t}]$
> is tighter than the gradient mapping measure $\mathbb{E}|| \mathcal{G}_\mathcal{X} (x_t,\nabla f(x_t),\gamma) ||$.
>
> In fact, we can also use a new convergence measure $\mathcal{M}_t^+ = \frac{1}{\rho} \mathcal{M}_t$ to obtain the same gradient complexities, except that we only choose different parameters $G$ and $M$ in Theorems 1 and 2, respectively.
> Under this case, this measure $\mathbb{E}[ \mathcal{M}_t^+ ] $  is always tighter than the gradient mapping measure $\mathbb{E}|| \mathcal{G}_\mathcal{X} (x_t,\nabla f(x_t),\gamma) ||$.  In addition, I think that the new metric $\mathbb{E}[ \mathcal{M}_t^+ ] $ also is more effective than the metric  $\mathbb{E}[ \mathcal{M}_t ] $ in unconstrained adaptive case.
> In our final version, we will use this convergence measure $\mathcal{M}_t^+$ instead of $\mathcal{M}_t$.
>
> 2) In fact, different problems generally rely on different specific adaptive learning rates.  In the numerical experiments, we only use the adaptive matrix $H_t$ given in the case 1. Although our algorithm uses the same adaptive matrix form in the experiments, we use different tuning parameters of adaptive matrix $H_t$ in different tasks.
>
> 3) Yes, we use the same adaptive learning rates in super-adam $(\tau=1)$ and super-adam $(\tau=0)$.
>
> Thank you very much for pointing these typos. In our final version, we will correct them.

---

### Official Review · Reviewer_Tq1x · 2021-07-16

**Rating:** 8
**Confidence:** 5

**Summary:**

This paper proposes a faster and universal framework of adaptive gradients (super-adam) for nonconvex stochastic optimization. Specifically, super-adam introduces a universal adaptive matrix that includes most existing adaptive gradient forms and many new adaptive learning rates. In particular, it can flexibly integrate the momentum and variance reduced techniques. Moreover, this paper provides an effective and interesting theoretical analysis framework based on a new convergence metric. Meanwhile, it provides the extensive experimental results to demonstrate the efficiency of the super-adam algorithm. Overall, it is an interesting and innovative paper. It will have a wide range of applications in machine learning.

**Ethics Review Area:**

["I don’t know"]

**Limitations And Societal Impact:**

See above

**Main Review:**

Some comments:

1.  I recommend the authors to proofread the paper again.
Some typos:
1) In the line 9: “our framework can flexibly integrates …” should be “our framework can flexibly integrate…”
2) In the line 115: “proposed to adapt the stepsize” should be “proposed to adopt the stepsize”
3) At the step 13 of super-adam algorithm, there is a symbol error: “[g_t - ” should be “[ g_t + ”.
In the line 436, the gradient estimator g_{t+1} is right. So I think that this symbol error at the step 13 of algorithm is a typo.
4) In the line 145, “g_t - \nabla f(x_{t+1};\xi_{t+1})” should be “g_t - \nabla f(x_t;\xi_{t+1})”.

2.  I also recommend the authors to detail the adaptive learning rates used in the super-adam algorithm in the experiment.

----After Rebuttal----
As the author's rebuttal well addressed my main concerns, I increased my score to 8 to support this paper. Thanks.

**Time Spent Reviewing:**

3

---

> ### Author Response · Authors · 2021-08-09
> **Responses for comments**
>
> Thanks for your positive comments. We will proofread our paper again. In our final version, we will correct these typos. In the experiment, we use the adaptive matrix $H_t$ given at the case 1 in our Super-Adam algorithm.

---

### Official Review · Reviewer_PnCw · 2021-07-16

**Rating:** 4
**Confidence:** 4

**Summary:**

This manuscript introduces a unified framework (i.e., SUPER-ADAM) of adaptive gradient methods for nonconvex optimization. It is proved that a universal convergence measure converges with a fast rate, which recovers the $O(\epsilon^{-3})$ complexity by STORM. Another result is that it also recovers $O(\epsilon^{-4})$ result by SGD, when individual function does not have a Lipschitz-continuous gradient. Numerical experiments on image classification and language modeling shows that SUPER-ADAM has faster convergence rate when training deep neural networks.

**Limitations And Societal Impact:**

Yes.

**Main Review:**

The unified framework is new to me, but the only algorithmic innovation is line 12 in Algorithm 1. The choices of $\tau=0$ or $\tau=1$ are nothing but hyper-parameters which make the algorithm use the standard gradient estimator or the STORM estimator. I have the following concerns.

1. Convergence Measure. The authors claim that the proposed measure in (15) is tighter than existing measures. I agree that it is the case when we are facing unconstrained nonconvex optimization with non-coordinate-wise adaptive learning rate. What about constrained case and adaptive case in which the preconditioning matrix is not identity? I do not think it is still tighter. For example, the gradient mapping may not be a lower bound for the measure in (15). In unconstrained adaptive learning rate case, $\rho H_t^{-1}$ should be very small when $t$ gets large, so it is also not an upper bound of the gradient norm. For fair comparison, is it possible to prove the convergence measure in the usual sense (i.e., gradient mapping in constrained case and gradient norm in unconstrained adaptive case)?

2. As I mentioned, line 12 is the only difference compared with existing methods. Why line 12 is important to recover the convergence rate of STORM? There is no doubt that in unconstrained case, without line 12 and with $\tau=1$, the algorithm is the same as STORM and hence can recover $O(\epsilon^{-3})$ rate. Also I would like to ask the same question in terms of SGD.

3. After inspecting Lemma 2 and Lemma 3 and Theorem 3 in this manuscript, I think the proofs are almost adapted from STORM paper, except for plugging in the Lemma 1 in [10] and the line 11 of Algorithm 1. Please clarify what are the technical innovations when compared with Lemma 1, Lemma 2 and Theorem 1 in the STORM paper.

Experiments:
For SUPER-ADAM with $\tau=1$, since the algorithm need to calculate two stochastic gradients at one iteration, so the running time might be slow. Is it possible to report the running time results?

From my point of view, the theoretical contribution is minimal for unconstrained problems, since SGD and STORM are already optimal with different assumptions, and SUPER-ADAM's algorithm design is very similar to them except for the line 12. The convergence measure when using coordinate-wise adaptive learning rate for unconstrained problems is not comparable with the usual gradient norm.

SUPER-ADAM might be interesting in constrained case. However, the convergence measure is not comparable with gradient mapping, and there are no empirical studies for constrained problems in this submission.

Due to these reasons, it is hard to say that the algorithm is universal.

**Time Spent Reviewing:**

3 hours

---

> ### Author Response · Authors · 2021-08-09
> **Responses for comments**
>
> Thanks for your comment and suggestion. We address your concerns one by one as follows:
>
> 1) We first declare that our new unified framework for adaptive gradients has been proposed by introducing a universal adaptive matrix, which can use almost all existing adaptive learning rates (including both global and coordinate-wise learning rates) and some new adaptive learning rates such as Barzilai-Borwein type adaptive learning rate given at (9) in our paper. Moreover, our framework can flexibly integrate the momentum and variance-reduced techniques. For example, when $\tau=1$, our algorithm use the variance reduced technique of STORM. When $\tau=0$, our algorithm use the basic momentum technique as in Adam algorithm.
>
> 2) We admit that our new convergence measure $\mathcal{M}_t$ is not tighter than the existing measure $\mathbb{E}||\nabla f(x_t)||$ for unconstrained optimization when $ H_t\neq I_d$. However, our convergence measure is still effective. Although some cases can not ensure $\rho H^{-1}_t \rightarrow I_d$, when $\mathcal{M}_t\rightarrow 0$, we still obtain $||\nabla f(x_t)|| \rightarrow 0$. Specifically, due to $H^{-1}_t\succ 0$, when $ \mathcal{M}_t = ||\nabla f(x_t)-g_t|| + \rho||H^{-1}_t g_t|| \rightarrow 0$, we have $\nabla f(x_t)\rightarrow g_t$ and $g_t \rightarrow 0 $. Then we have $\nabla f(x_t)\rightarrow 0$.
>
> We need to argue that $\rho H^{-1}_t$ is not very small when $T$ gets large in many cases. For example, when we use the new Barzilai-Borwein type adaptive learning rate given at (9) in our paper, clearly, we have $\rho H^{-1}_t \geq \frac{\rho}{L+\lambda} $. We also argue that most of adaptive gradient methods assume the stochastic gradient is bounded and sparse (Please see Table 1 in our paper). For example, when we use the existing adaptive learning rate such as the case 1 in our algorithm, clearly, $\rho H^{-1}_t $ is not very small when $T$ gets large due to the sparse and bounded stochastic gradients.  Note that although Table 1 shows that Adaptive SGD [15] does not assume the bounded stochastic gradients, it assumes that the function $f(x)$ is $L$-Lipschitz and the noise in the stochastic gradient $ g(x,\xi)$ has bounded support (Please see H2 and H4 at page 3 in http://proceedings.mlr.press/v89/li19c/li19c.pdf). In the other words, Adaptive SGD actually also assumes the bounded stochastic gradients. Since $f(x)$ is $L$-Lipschitz, we have $||\nabla f(x)||\leq L $. Due to the bounded noise in the stochastic gradient $ g(x,\xi)$ (i.e., $||g(x,\xi)- \nabla f(x)||\leq S$), we have $|| g(x,\xi)|| \leq || \nabla f(x)|| + || g(x,\xi)- \nabla f(x)|| \leq L +S$, i.e., the bounded stochastic gradients.
>
> Thanks for your suggestion. In our final version, we will provide the convergence results based on  gradient mapping in constrained case and gradient norm in unconstrained adaptive case, respectively.
>
> Note that for the constrained case, please see the following response 6.
>
> 3) We admit that the proof of our Theorem 1 follows the proof of non-adaptive version of STORM algorithm [7] (i.e, Theorem 2 at page 14 in [7a]). In fact, the novelty of our convergence analysis mainly lies in our Lemma 2. In our Lemma 2, we first use a mirror descent iteration instead of the step 11 in our algorithm, and then cleverly use Lemma 1 [10] to obtain a key result in our convergence analysis. While Lemma 1 in [7] only applies the smoothness of objective function to obtain a bound for the decline of the objective function, which is commonly used in convergence analysis of nonconvex optimization. Clearly, there exist some significance differences between our Lemma 2 and Lemma 1 in [7]. Since we use the same variance reduced technique as in the STORM algorithm, our Lemma 3 basically follows the Lemma 2 in [7].
> In our final version, we will point the common and differences between our convergence analysis and the convergence analysis of STORM algorithm.
>
> [7a] Momentum-Based Variance Reduction in Non-Convex SGD, https://arxiv.org/pdf/1905.10018.pdf
>
> Due to different bounded variances of stochastic gradients (please see Lemma 3 and Lemma 4 in our paper), there exist some significant  differences between the proofs of our Theorem 1 and Theorem 2. For example, we choose different parameters $\mu_t$ and $ \alpha_t$ and establish different Lyapunov functions $ \Phi_t$ and $ \Omega_t$. Specifically, the Lyapunov function $ \Phi_t$ relies on the dynamic parameter $\mu_t$, while $ \Omega_t$ does not rely on the dynamic parameter $\mu_t$. Thus, the proof of our Theorem 2 can not easily follow the proof of STORM algorithm [7].
>
> 4) To the unconstrained optimization, clearly, there exist some significance differences between our Super-Adam algorithm and SGD, STORM algorithms. When $\tau=0$, our super-Adam algorithm can use different adaptive learning rates at the step 11 in our algorithm 1, and use the momentum-based stochastic gradient at the step 13. While SGD only use a simple stochastic gradient descent based on a mini-batch stochastic gradient.  When $\tau=1$, although our Super-Adam algorithm uses the same variance reduced technique as in the STORM algorithm, our super-Adam algorithm can use different adaptive learning rates including the existing global and coordinate-wise learning rates. While STORM algorithm only uses a specific decreasing global learning rate.
>
> 5) Yes, you are right. For the case $\tau=1$, our method use more time to evaluate one more gradient. Take the CIFAR-10 as an example, for optimizers like Adam, it takes around 33s-35s to finish one epoch, while our method takes around 53s-55s. In our final version, we will add the results of loss vs. time.
>
> 6) We need to declare that our convergence measure $\mathcal{M}_{t}$ defined in (15) in our paper is comparable with gradient mapping under the constrained optimization. When $\rho \geq 1$, our convergence measure $\mathcal{M}_t$ is tighter than the gradient mapping measure $||\mathcal{G}_\mathcal{X} (x_t,\nabla f(x_t),\gamma) ||$ (Please see lines 193-199 at page 6).
>
> Specifically, according to Proposition 1 in [10], we have $||\mathcal{G}_\mathcal{X}(x_t,g_t,\gamma)-\mathcal{G}_\mathcal{X} (x_t,\nabla f(x_t),\gamma)|| \leq \frac{1}{\rho} || \nabla f(x_t)-g_t ||$.
>  Since $||\mathcal{G}_\mathcal{X}(x_t,\nabla f(x_t),\gamma)||  \leq  ||\mathcal{G}_\mathcal{X}(x_t,g_t,\gamma)|| + ||\mathcal{G}_\mathcal{X}(x_t,g_t,\gamma)-\mathcal{G}_\mathcal{X}(x_t,\nabla f(x_t),\gamma)||$,
>  we have $||\mathcal{G}_\mathcal{X}(x_t,\nabla f(x_t),\gamma)|| \leq ||\mathcal{G}_\mathcal{X}(x_t,g_t,\gamma)|| + \frac{1}{\rho}||\nabla f(x_t)-g_t|| = \frac{1}{\rho}(  \rho ||\mathcal{G}_\mathcal{X} (x_t,g_t,\gamma)|| + ||\nabla f(x_t) - g_t|| ) = \frac{1}{\rho} \mathcal{M}_t $.
>
> When $\mathcal{M}_{t}   \rightarrow 0$,
> we can obtain $|| \mathcal{G}_\mathcal{X} (x_t,\nabla f(x_t),\gamma) || \rightarrow 0$, where $x_t$ is a stationary point or local minimum of the problem (1).
>
> Clearly, when $\rho \geq 1$, our measure $\mathbb{E}[\mathcal{M}_{t}]$
> is tighter than the gradient mapping measure $\mathbb{E}|| \mathcal{G}_\mathcal{X} (x_t,\nabla f(x_t),\gamma) ||$.
>
> In fact, we can also use a new convergence measure $\mathcal{M}_t^+ = \frac{1}{\rho} \mathcal{M}_t$ to obtain the same gradient complexities, except that we only choose different parameters $G$ and $M$ in Theorems 1 and 2, respectively.
> Under this case, this measure $\mathbb{E}[ \mathcal{M}_t^+ ] $  is always tighter than the gradient mapping measure $\mathbb{E}|| \mathcal{G}_\mathcal{X} (x_t,\nabla f(x_t),\gamma) ||$. In addition, I think that the new metric $\mathbb{E}[ \mathcal{M}_t^+ ] $ also is more effective than the metric  $\mathbb{E}[ \mathcal{M}_t ] $ in unconstrained adaptive case.
> In our final version, we will use this convergence measure $\mathcal{M}_t^+$ instead of $\mathcal{M}_t$.
>
> 7) Thanks for your suggestion. We train CIFAR-10 over ResNet-18 with a $L_1$ constraint, which encourages the sparsity of the neural network weights, i.e., $\min_x \mathbb{E}[f(x;\xi)], \ s.t.\ ||x||_1\leq \varepsilon$. The results (training loss) are summarized in the following table ($\varepsilon=10^{3}$). From the following results, our Super-Adam outperforms the Adam with a great margin. In our final version, we will add more experiment results on the constrained optimization.
> |Optimizer|10 epochs|20 epochs|50 epochs|100 epochs|150 epochs|
> |---|---|---|---|---|---|
> |Adam|1.184|1.059|0.980|0.944|0.920
> |Super-Adam ($\tau=0$)| 1.104 | 0.982 | 0.853 | 0.780 | 0.739|

---

### Official Review · Reviewer_eztp · 2021-07-18

**Rating:** 4
**Confidence:** 5

**Summary:**

This paper proposes a universal framework for adaptive gradients methods by introducing a universal adaptive matrix. It also offers the theoretical convergence analysis for this optimizing method, and proves that this method can achieve the complexity of $O(\epsilon^{-3})$ for finding an $\epsilon$-stationary point.

**Limitations And Societal Impact:**

The author need to  address the problesm above.

**Main Review:**

The main concern is about the practicability of SUPER-Adam in the Case 2 in Algorithm 1. It is known that the number of parameters of modern DNNs is typically more than 10 million, such as the number of parameters of VGG16 is ~134 million and that of ResNet18 is 33 million. Therefore, $H_t$ in SUPER-Adam in Case 2 will be at least 8 order of magnitude larger than that in $H_t$ in SUPER-Adam in Case 1 that is commonly used, so we need to scale the learning rate $\eta_t$ by a factor of $>10$ million to maintain the comparable convergence rate with Adam. This overlarge factor may be impractical. Actually, from Section 6.1, we know the authors also did not set $\eta$ so large.

When $M_t \rightarrow 0$ , SUPER-Adam in the Case 1 cannot ensure $\rho H_t^{-1} \rightarrow I_d$ in Eq. (17), and then $\Vert\nabla f(x_t) \Vert \rightarrow 0$ will not hold. In this case, Theorem 1 and Theorem 2 might be somewhat meaningless.


Theorem 1 and Theorem 2 and their proofs are obviously followed the paper that proposed STORM with a little change, but the authors did not report this close relation. Therefore, the bright spot of the convergence analysis is discounted

From the 1st and 5th subfigure of figure 2, we know the learning rate for SUPER-Adam decays by a factor (maybe 10) at the 70th and 100th epoch, respectively, which is not described in experimental settings. Moreover, this learning rate decay strategy seems to be not used for other optimizing methods except AdaBlief. Hence, the comparison may be unfair. Additionally, it is known that SGD-type methods commonly perform better for image classification than adaptive gradient methods, so I would like to see SGD will be also compared to SUPER-Adam.



**Time Spent Reviewing:**

10 hours

---

> ### Author Response · Authors · 2021-08-09
> **Responses for comments**
>
> Thanks for your comments. We address your concerns one by one as follows:
>
> 1) We provide the case 2 in our algorithm, which only is a global adaptive matrix example as the global adaptive learning rate used in AdaGrad-Norm [24]. In the numerical experiments, we only use the adaptive matrix $H_t$ given in the case 1. In fact, we can gave a reasonable and novel global adaptive learning rate: given $\beta\in (0,1)$, $b_t=\beta b_{t-1} + (1-\beta)|| \nabla f(x_t;\xi_t)||$ and $H_t = (b_t + \lambda)I_d$. In the experiments, this global adaptive learning rate works well.
>
> 2) Although some cases can not ensure $\rho H^{-1}_t \rightarrow I_d$, we still can obtain $||\nabla f(x_t)|| \rightarrow 0$ when $\mathcal{M}_t\rightarrow 0$. Specifically, due to $H^{-1}_t \succ 0$, when $ \mathcal{M}_t = ||\nabla f(x_t)-g_t|| + \rho||H^{-1}_t g_t|| \rightarrow 0$, we have $\nabla f(x_t)\rightarrow g_t$ and $g_t \rightarrow 0$. Then we have $\nabla f(x_t)\rightarrow 0$.
>
> 3) We admit that the proof of Theorem 1 follows the proof of non-adaptive version of STORM algorithm [7]. In our final version, we will detail the common and differences between our convergence analysis and the convergence analysis of STORM algorithm. In fact, the novelty of our convergence analysis mainly lies in our Lemma 2. In Lemma 2, we first use a mirror descent iteration instead of the step 11 in our algorithm, and then cleverly use Lemma 1 [10] to obtain a key result in our convergence analysis.
> Due to different bounded variances of stochastic gradients (please see Lemma 3 and Lemma 4), there exist some significant differences between our proofs of Theorem 1 and Theorem 2. For example, we choose different parameters $\mu_t$ and $\alpha_t$ and establish different Lyapunov functions $ \Phi_t$ and $ \Omega_t$. Specifically, the Lyapunov function $ \Phi_t$ relies on the dynamic parameter $\mu_t$, while $ \Omega_t$ does not rely on the dynamic parameter $\mu_t$. Thus, the proof of Theorem 2 can not follow the proof of STORM algorithm [7].
>
> 4) Thank you for pointing this learning rate schedule issue. In the experiment, Adam, AmsGrad, AdaGrad-Norm and Adam$^+$  do not decay the learing rate, while our Super-Adam and AdaBelief shrink the learning rate by 10 at 100 and 150 epochs. For fair comparison, we add the same decay factors for the remained optimizers. The results (training losses) on CIFAR-100 are summarized in the following table. From the following results, we find that our Super-Adam still outperforms other optimizers.
> |$\textit{Optimizer}$ | 10 epochs | 20 epochs | 50 epochs | 100 epochs | 150 epochs |
> | --- | --- | --- | --- |--- | --- |
>  |AdaGrad-Norm | 2.102  | 1.258 | 0.278 | 0.035 | 0.038 |
>  |Adam   | 2.305   |  1.460 | 0.541 | 0.196 | 0.059|
>  |AmsGrad  | 2.402  | 1.444 | 0.461 | 0.132 | 0.024|
>  |Storm |  2.206 | 1.400 | 0.578 | 0.177 | 0.053|
>  |AdaBelief | 2.444 | 1.686 | 0.568 | 0.166 | 0.031|
>  |Adam$^+$ |  1.767 | 1.088 | 0.190 | 0.048 | 0.008|
>  Super-Adam ($\tau=0$) | 1.789 | 1.187 | 0.416 | 0.116 | 0.060|
>  Super-Adam ($\tau=1$) | 1.544 | 0.905 | 0.162 | 0.027 | 0.005|
> 5) Our Super-Adam algorithm achieves comparable or even better results than the SGD. For example, we got 93.58% accuracy ($\tau=1$) and 93.18% accuracy ($\tau=0$) on CIFAR-10 compared to the 93.38% for SGD. In our final version, we will add the SGD in all experiments.

---

### Decision · Program_Chairs · 2021-09-27

**Decision:**

Accept (Poster)

**Comment:**

During the discussion phase, the paper has received intensive discussion between the authors and the reviewers about the merits and the concerns. Two reviewers have a strong favor of accepting this paper given that the paper presents a new analysis of Adam method for constrained non-convex optimization and also an improved variant of Adam by using recursive variance reduction methods with an optimal complexity. The major concerns are from the used convergence measure that is different from the standard convergence measure, and the authors' argument about its implication for the convergence of the standard  measure in terms of (proximal) gradient norm. The AC agrees that the presented results are interesting and the analysis of the Adam-style methods for constrained non-convex optimization is novel. The concern about the inconsistency between the presented convergence measure and the standard convergence measure is understandable given that the paper considers the constrained non-convex optimization. However, the authors should weaken their argument about its implication for the convergence of the gradient  norm or add more evidence such as empirical results presented in the rebuttal for further supporting their claim. The AC believes the concern should be addressable, and hence recommends an acceptance.